# Deep Principal Support Vector Machines
# for Nonlinear Sufficient Dimension Reduction

**YinFeng Chen** [* 1]   **Jin Liu** [2]   **Rui Qiu** [* 3]

## Abstract

The normal vectors obtained from the support vector machine (SVM) method offer the potential to achieve sufficient dimension reduction in both classification and regression scenarios. Motivated by it, we in this paper introduce a unified framework for nonlinear sufficient dimension reduction based on classification ensemble. Kernel principal SVM, which leverages the reproducing kernel Hilbert space, can almost be regarded as a special case of this framework, and we generalize it by using a neural network function class for more flexible deep nonlinear reduction. We theoretically prove its unbiasedness with respect to the central $\sigma$-field and provide a nonasymptotic upper bound for the estimation error. Simulations and real data analysis demonstrate the considerable competitiveness of the proposed method, especially under heavy data contamination, large sample sizes, and complex inputs.

## 1. Introduction

With the advent of the big data era, high-dimensional data have become increasingly common across diverse scientific domains. To this end, sufficient dimension reduction (SDR) techniques are developed to extract low-dimensional representations of data while preserving as much information as possible. These representations serve to facilitate both visualization and downstream tasks of data analysis. Let $X \in \mathbb{R}^p$ be the predictor variables of dimension $p$, and $Y \in \mathbb{R}$ be the response variable. The classical linear SDR method aims to find a $p \times d(d < p)$ matrix $\eta$ such that the following conditional independence holds:

$$Y \perp\!\!\!\perp X | \eta^T X, \tag{1}$$

where $\perp\!\!\!\perp$ stands for statistical independence. The identifiable parameter in (1) is the subspace spanned by the columns of $\eta$, rather than $\eta$ itself, referred to as the sufficient dimension reduction subspace. Under mild conditions (see Cook (1998); Yin et al. (2008)), the intersection of all such spaces is also a sufficient dimension reduction subspace and is called the central subspace, denoted by $\mathscr{S}_{Y|X}$. Representative methods of linear SDR include sliced inverse regression (SIR) (Li, 1991), sliced average variance estimate (SAVE) (Cook & Weisberg, 1991), directional regression (DR) (Li & Wang, 2007), minimum average variance estimation (MAVE) (Xia et al., 2002), and so on. We refer to Ma & Zhu (2013) for an extensive survey on linear SDR.

Sometimes due to complex regression relationships, linear combinations of predictor variables often fail to achieve effective SDR. To address this limitation, a nonlinear extension of the linear SDR, as formulated in Cook (2007), is considered to seek any function $\boldsymbol{f}(X) : \mathbb{R}^p \mapsto \mathbb{R}^d$ such that

$$Y \perp\!\!\!\perp X | \boldsymbol{f}(X). \tag{2}$$

The kernel trick stands out as the most popular approach to extend linear SDR to nonlinear SDR, and related works include Wu (2008), Yeh et al. (2009), among others. Recently, Lee et al. (2013) proposed a rigorous SDR framework based on $\sigma$-fileds, which integrates both linear and nonlinear SDR into a unified system. They define the SDR problem as seeking a sub $\sigma$-field $\mathscr{G}$ of $\sigma(X)$ such that

$$Y \perp\!\!\!\perp X | \mathscr{G}. \tag{3}$$

Similar to the linear scenario, the intersection of all sub $\sigma$-fields satisfying relation (3) still conforms to (3) under mild conditions. This minimal sub $\sigma$-field is then called the central $\sigma$-field and denoted by $\mathscr{G}_{Y|X}$. Definitions of (1) and (2) can be seamlessly integrated into this framework by considering $\mathscr{G}$ as $\sigma(\eta^T X)$ and $\sigma(\boldsymbol{f}(X))$, respectively. Compared to the linear case, the development of nonlinear SDR methods is relatively limited. As the nonlinear counterparts of SIR and SAVE, Lee et al. (2013) introduced

---

*Equal contribution  [1]School of Mathematics and Statistics, Xi'an Jiaotong University, Xi'an, China [2]School of Data Science, The Chinese University of Hong Kong (Shenzhen), Shenzhen, China [3]School of Mathematical Sciences, Peking University, Beijing, China. Correspondence to: Rui Qiu <rqiu@pku.edu.cn>.

*Proceedings of the 42$^{nd}$ International Conference on Machine Learning*, Vancouver, Canada. PMLR 267, 2025. Copyright 2025 by the author(s).

the generalized sliced inverse regression (GSIR) and generalized sliced average variance estimator (GSAVE). These methods may struggle with large samples and complex input structures.

The rapid development of machine learning has sparked interest in incorporating machine learning techniques into traditional dimension reduction, such as k-means inverse regression (Setodji & Cook, 2004), principal support vector machines (Li et al., 2011), and dimension reduction forests (Loyal et al., 2022). In particular, deep neural networks have attracted the attention of many researchers due to their powerful feature extraction capabilities. A prominent representative of deep unsupervised dimension reduction is the auto-encoder models with bottleneck structures in representation learning (Hinton & Salakhutdinov, 2006). Inspired by the mechanism of generative adversarial networks (GANs), Zheng et al. (2022) trained two deep neural networks adversarially and proposed a mutual information-based sufficient representation learning approach. Further, to avoid estimating the ratio of the joint density and the marginal densities, Huang et al. (2024) replaced mutual information with distance covariance and established the consistency of the proposal but not the rate of convergence. However, their strategy for limiting the distribution of sufficient representation distinguished itself from the central $\sigma$-filed framework used in our paper. Recently, Chen et al. (2024) proposed GMDDNet for nonlinear SDR, which combines the generalized martingale difference divergence criteria with neural networks. In addition, Kapla et al. (2022) introduced a two-stage SDR estimation utilizing a neural network instead of local linear smoothing typically employed in MAVE (Xia et al., 2002). However, it remains a linear method.

This paper will introduce a new nonlinear SDR method based on classification ensembles within the standard statistical framework established by (3). Our contribution lies in the following aspects:

1. Our nonlinear SDR methods can be applied to a broad range of function classes. Among them, the neural network approximator shows significant promise. This deep method demonstrates flexibility in handling large samples and diverse input types including images and text. Importantly, it is robust for regression and classification tasks, particularly suited for data with many outliers. Our deep method does not require additional special design of network structures such as bottleneck and generative adversarial structures.

2. We rigorously establish the unbiasedness of optimal solutions to the proposed objective function, indicating that they belong to the central class. Under the assumption concerning the covering number of function classes, the upper bound of the estimation error

is given. Encouragingly, the assumption can be satisfied by Vapnik-Chervonenkis (VC) classes, with neural networks being a representative example. As a result, we derive a specific convergence rate tailored to the neural network class, showcasing its optimality up to a polynomial order of $\log n$ when compared against the minimax rate of nonparametric regression.

3. The structure order determination method from classical sufficient dimension reduction can be easily applied to our method, which is not the case for other neural network-based methods.

**Notations:** For any vector $x$, $\|x\|_r$ and $\|x\|_\infty$ represent the $l_r$ and supremum norm of $x$, respectively. Let $L_2(P_X)$ be the space of square-integrable measurable functions with respect to the measure $P_X$ of $X$. For any function $f \in L_2(P_X)$, we define $\|f\|_{L_2(P_X)} = \left\{ E|f(X)|^2 \right\}^{1/2}$ and $\|f\|_\infty = \sup_x |f(x)|$ as the $L_2(P_X)$ norm and supremum norm of $f$, respectively. The $\epsilon$-covering number of a set $A$ with respect to the metric $d$ is denoted by $N(\epsilon, A, d)$.

## 2. Preliminaries

To better elucidate our nonlinear SDR method and related theories, we first provide a detailed introduction to the framework of nonlinear SDR. Deep nonlinear SDR is the focus of this paper, for which we will briefly describe the structure of neural networks.

### 2.1. Nonlinear SDR

Let $(\Omega, \mathcal{F}, P)$ denote a probability space, and let $(\Omega_X, \mathscr{B}_X)$, $(\Omega_Y, \mathscr{B}_Y)$ and $(\Omega_{XY}, \mathscr{B}_{XY})$ represent measurable spaces where $\Omega_X \subset \mathbb{R}^p$, $\Omega_Y \subset \mathbb{R}^q$ and $\Omega_{XY} \subset \mathbb{R}^{p+q}$. By default, $\Omega_{XY} = \Omega_X \times \Omega_Y$ and $\mathscr{B}_{XY} = \mathscr{B}_X \times \mathscr{B}_Y$. Let $X$, $Y$, and $(X, Y)$ be random vectors taking values in $\Omega_X$, $\Omega_Y$, and $\Omega_{XY}$, respectively, with distributions $P_X$, $P_Y$, and $P_{XY}$. The $\sigma$-filed generated by $X$ is $\sigma(X) = X^{-1}(\mathscr{B}_X)$. The conditional distribution of $X$ given $Y$ is denoted by $P_{X|Y}(\cdot|\cdot) : \mathscr{B}_X \times \Omega_Y \to \mathbb{R}$. We consider the following nonlinear SDR problem

$$Y \perp\!\!\!\perp X | \boldsymbol{f}_0(X), \tag{4}$$

where $\boldsymbol{f}_0(\cdot)$ is a $\mathbb{R}^d$-valued function ($d < p$). That is to say, at least $d$ functions of $p$-dimensional $X$ are needed to fully capture the relationship between $Y$ and $X$. Here, we assume the structural dimension $d$ is known for simplicity. Obviously, $\boldsymbol{f}_0$ is unidentifiable since we can substitute $\boldsymbol{f}_0$ with any one-to-one function of it, without altering the conditional independence (4). In fact, this conditional independence relies solely on the $\sigma$-field generated by $\boldsymbol{f}_0(X)$, which is unique under very mild assumptions (Lee et al., 2013). Without loss of generality, we may always assume

**Assumption.** The family of probability measures $\{P_{X|Y}(\cdot|y) : y \in \Omega\}$ is dominated by a $\sigma$-finite measure. Then there exists a unique minimal sufficient $\sigma$-field (or central $\sigma$-field) $\mathscr{G}_{Y|X}$ such that

$$Y \perp\!\!\!\perp X | \mathscr{G}_{Y|X}.$$

Moreover, $\boldsymbol{f}_0$ in (4) satisfies

$$E(\boldsymbol{f}_0(X)) = 0, \text{Var}(\boldsymbol{f}_0(X)) = I_d,$$

and

$$\sigma(\boldsymbol{f}_0(X)) = \mathscr{G}_{Y|X}.$$

We introduce some examples to build intuition in Appendix A.1. Even though the central $\sigma$-field is identifiable, it can be quite abstract to handle. Following Lee et al. (2013), let $\mathfrak{M}_{Y|X}$ be the central class corresponding to the central $\sigma$-field $\sigma(\boldsymbol{f}_0(X))$. This class is the collection of all square-integrable and $\sigma(\boldsymbol{f}_0(X))$-measurable functions. Equivalently, $\mathfrak{M}_{Y|X}$ is a subset of $L_2(P_X)$ containing all elements that can be expressed as a function of $\boldsymbol{f}_0$. To introduce the concept of unbiasedness for nonlinear SDR, we first review the unbiasedness in linear SDR. The central subspace $\mathscr{S}_{Y|X} = \text{Span}(\eta)$ can be viewed as a member of the parameter space that encompasses all $d$-dimensional subspaces of $\mathbb{R}^p$. Let $P_n$ be the empirical distribution of an i.i.d. sample $\{(X_1, Y_1), \ldots, (X_n, Y_n)\}$ from the distribution $P_{XY}$. $T$ is a mapping from all possible distributions of $(X, Y)$ to $\mathbb{R}^{p \times d}$. If $\text{Span}[T(P_{XY})] \subseteq \mathscr{S}_{Y|X}$, we call $T(P_n)$ is an unbiased estimator of $\mathscr{S}_{Y|X}$. This is equivalent to the inner product of $X$ and any column vector of $T(P_{XY})$ being measurable with respect to $\sigma(\eta^T X)$. From this perspective, we can easily generalize the concept of unbiasedness to nonlinear SDR:

**Definition 2.1.** A function $f(X)$ from $L_2(P_X)$ is unbiased for the central $\sigma$-field $\mathscr{G}_{Y|X}$ if $f(X) \in \mathfrak{M}_{Y|X}$. Such a function is called an SDR function.

Unbiasedness is the most fundamental requirement for SDR estimators. It ensures that the information extracted by each SDR function solely originates from the regression relationship between $X$ and $Y$.

### 2.2. Deep neural network

We now briefly introduce the feedforward neural networks with the rectified linear unit (ReLU) activation function. Specifically, the feedforward neural network can be represented as

$$f(x) = W_{\mathcal{L}} \circ \sigma \circ W_{\mathcal{L}-1} \circ \sigma \circ \cdots \circ \sigma \circ W_1 \circ \sigma \circ W_0(x),$$

where $x \in \mathbb{R}^p$, $W_i(x) = \omega_i x + b_i$ denotes an affine transformation with the weight matrix $\omega_i \in \mathbb{R}^{k_{i+1} \times k_i}$ and intercept vector $b_i \in \mathbb{R}^{k_{i+1}}$, for $i = 0, 1, \ldots, \mathcal{L}$. The function

$\sigma(x) = \max\{x, 0\}$ is the ReLU activation function applied element-wise. Here, $k_0$ and $k_{\mathcal{L}+1}$ are the input and output dimensions, respectively. And $k_i, i = 1, 2, \ldots, \mathcal{L}$ stands for the number of neurons (width) in the $i$th hidden layer. The depth of a network, denoted by $\mathcal{L}$, refers to the number of hidden layers, while its width denotes the maximum width, $\mathcal{N} = \max\{k_1, k_2, \ldots, k_{\mathcal{L}}\}$, among the hidden layers. Moreover, the size of a network can be further characterized by two quantities: the total number of neurons $\mathcal{U} = \sum_{i=1}^{\mathcal{L}} k_i$, and the total number of parameters $\mathcal{S} = \sum_{i=0}^{\mathcal{L}} k_{i+1} \cdot (k_i + 1)$. As the most widely used function class, neural networks will serve as the main estimator for the nonlinear SDR functions in our methods.

## 3. Motivation and Methodology

In this section, we introduce a new nonlinear SDR method based on the principal support vector machines (PSVM) proposed by Li et al. (2011). Then we give a reliable strategy for determining the structure dimension $d$ in practical applications.

### 3.1. Sample estimation for nonlinear SDR

When the response $Y \in \{-1, 1\}$ is binary, Li et al. (2011) observed that the normal vectors of the hyperplane learned by support vector machines (SVM) align with the directions forming the SDR subspace. Therefore these normal vectors can be naturally leveraged to construct the SDR subspace. Specifically, given training data $\mathscr{D}_n = \{(X_i, Y_i)\}_{i=1}^n$, Li et al. (2011) demonstrated the validity of the following objective function for linear SDR

$$\min_{(\psi,t) \in \mathbb{R}^p \times \mathbb{R}} \frac{\lambda}{n} \sum_{i=1}^n \left[1 - Y_i\{\psi^\top(X_i - E_n(X)) - t\}\right]^+ + \text{Var}_n(\psi^\top X), \tag{5}$$

where $\lambda$ is a hyperparameter, $E_n(\cdot)$ and $\text{Var}_n(\cdot)$ denotes the empirical mean and variance, and $[x]^+ = \max\{x, 0\}$ is the hinge loss function. In particular, Li et al. (2011) proposed combining these normal vectors through principal component analysis to form the basis of the SDR subspace. Zhou & Zhu (2016) then introduced the principal minimax support vector machines formulation

$$\min_{(\psi,t) \in \mathbb{R}^p \times \mathbb{R}} \lambda\|\psi\|_1 + c^*\psi^T\psi + \text{Var}_n(\psi^\top X) + \frac{C}{n} \sum_{i=1}^n \left\{1 - Y_i\left(\psi^T X_i + t\right)\right\}^+,$$

where $\lambda, C$ are hyperparameters and $c^*$ is a special number. Additionally, Shin et al. (2017) proposed

$$\min_{(\psi,t) \in \mathbb{R}^p \times \mathbb{R}} \psi^\top\psi + \frac{\lambda}{n} \sum_{i=1}^n w_\pi(Y_i)\left\{1 - Y_i\left(\psi^\top X_i + t\right)\right\}^+,$$

where $\lambda$ is a hyperparameter and $w_\pi(Y_i) = 1 - \pi$ if $Y_i = 1$ and $\pi$ otherwise.

The kernel trick or reproducing kernel Hilbert space (RKHS) provides an effective approach for extending linear SDR methods to their nonlinear counterparts. However, finding a suitable kernel for non-vector input remains challenging. For example, when dealing with tensor-valued input such as images, the conventional kernel trick often requires flattening tensors, which inevitably results in information loss. In contrast, neural networks show remarkable flexibility in handling diverse data types. Moreover, the computational burden of neural networks is typically lower than that of the kernel trick when handling large samples. Given that the kernel trick frequently involves inverting an $n \times n$ kernel matrix, the computational cost $\mathcal{O}(n^3)$ becomes formidable for large $n$.

Motivated by the superiority of neural networks over the kernel trick and the validity of normal vectors from SVM in linear SDR, we combine (5) with a general functional class $\mathcal{F}_n$ (e.g., neural networks) and a general loss $\rho$ to achieve nonlinear SDR. Specifically, let $\widetilde{Y}$ be the binarization of $Y$, then we define an objective function

$$
\begin{aligned}
& L_n(f, t) \\
= & \lambda \operatorname{Var}_n(f(X)) + \lambda t^2 \\
& + \frac{1}{n} \sum_{i=1}^n \rho\big(-\widetilde{Y}_i\{f(X_i) - E_n(f(X)) - t\}\big) \quad (6) \\
= & \frac{2\lambda}{n(n-1)} \sum_{1 \le i < j \le n} \frac{\{f(X_i) - f(X_j)\}^2 + 2t^2}{2} \\
& + \frac{1}{n} \sum_{i=1}^n \rho\bigg(-\widetilde{Y}_i\bigg\{f(X_i) - \frac{1}{n}\sum_{j=1}^n f(X_j) - t\bigg\}\bigg),
\end{aligned}
$$

where $t \in \mathbb{R}$, $f$ is a member of $\mathcal{F}_n$ and $\rho$ is any non-decreasing convex loss function. Particularly, $\rho(x) = \max\{1 + x, 0\}$ corresponds to the hinge loss in (5). Here we follow the slicing techniques proposed in Li et al. (2011) to discretize $Y$ into the binary $\widetilde{Y}$. One, which we call "left versus right (LVR)", repeatedly divides the data $\mathscr{D}_n$ into two groups according to a set of cutting points for the response. The other, which we call "one versus another (OVA)", partitions $\mathscr{D}_n$ into several slices and pairs up all possible slices. Below we summarize the the algorithm process:

1. (LVR) Let $q_r, r = 1, \ldots, h - 1$, be $h - 1$ cutting points. For example, they can be equally spaced sample percentiles of $\{Y_1, \ldots, Y_n\}$. For $1 \le r \le h - 1$, set
$$
\widetilde{Y}_i^r = I(Y_i > q_r) - I(Y_i \le q_r).
$$
Over $\mathcal{F}_n \times \mathbb{R}$, $(\hat{f}_r, \hat{t}_r)$ is the minimizer of $L_n(f, t)$ with $\widetilde{Y}_i$ replaced by $\widetilde{Y}_i^r$.

1'. (OVA) In addition to the above $\{q_r\}_{r=1}^{h-1}$, let $q_0 = \min\{Y_1, \ldots, Y_n\}$ and $q_h = \max\{Y_1, \ldots, Y_n\}$. For each $(r, s)$ satisfying $1 \le r < s \le h$, set
$$
\widetilde{Y}_i^{rs} = I(q_{s-1} < Y_i \le q_s) - I(q_{r-1} < Y_i \le q_r).
$$
Over $\mathcal{F}_n \times \mathbb{R}$, $(\hat{f}_{rs}, \hat{t}_{rs})$ is the minimizer of $L_n(f, t)$ with $\widetilde{Y}_i$ replaced by $\widetilde{Y}_i^{rs}$.

2. Let $\mathbf{v}_1, \ldots, \mathbf{v}_d$ be the $d$ leading eigenvectors of either one of the matrices $M$ with entries
$$
M_{ij} = \sum_{k=1}^n \hat{f}_i(X_k)\hat{f}_j(X_k),
$$
where $\hat{f}_i$ is the minimizer from different dividing schemes ($1 \le i, j \le h - 1$ for LVR and $1 \le i, j \le h(h-1)/2$ for OVA). Then the $i$th ($1 \le i \le d$) nonlinear SDR function is
$$
\sum_j v_{ij}\hat{f}_j,
$$
where $v_{ij}$ is the $j$th entry of $l$-dimensional $\mathbf{v}_i$ ($l = h-1$ for LVR and $l = h(h-1)/2$ for OVA).

Li et al. (2011) empirically demonstrated that LVR works best when the response is a continuous variable and OVA works best when the response is categorical.

### 3.2. Estimation of structure dimension.

Determining the structural dimension $d$ is a critical aspect of SDR. Here we employ the ladle method proposed by Luo & Li (2016). This method utilizes not only the eigenvalue information of the matrix $M$ but also its eigenvectors to determine the structure dimension. Let $\mathcal{B}_k = (\mathbf{v}_1, \ldots, \mathbf{v}_k)$ and $(\lambda_1, \ldots, \lambda_k)$ respectively denote the $l \times k$ matrix comprising the first $k$ eigenvectors of $M$ and the vector comprising the first $k$ eigenvalues of $M$. Utilizing bootstrapping, we sample $n$ instances with replacement from a dataset of size $n$. With these $n$ samples, we re-implement $\mathcal{B}_k$, denoting its $i$th instance as $\mathcal{B}_{k,i}^*$. This bootstrapping procedure is repeated $n$ times to generate multiple bootstrap samples. To measure the disparity between $\mathcal{B}_k$ and its bootstrapped counterpart, we define the following function:

$$
f_n^0(k) = \begin{cases} 0, & k = 0, \\ n^{-1}\sum_{i=1}^n\{1 - |\det(\mathcal{B}_k^T \mathcal{B}_{k,i}^*)|\}, & k = 1, \ldots, l. \end{cases}
$$

We then normalize $f_n^0(k)$ as follows

$$
f_n(k) = \frac{f_n^0(k)}{1 + \sum_{i=0}^{r_l} f_n^0(i)},
$$

where $r_l = l - 1$ if $l \le 10$, $r_l = \lfloor l / \log l \rfloor$ if $l > 10$, and $\lfloor a \rfloor$ denotes the largest integer not exceeding $a$. The impact of eigenvalues is quantified by the function

$$g_n(k) = \frac{\lambda_{k+1}^2}{1 + \sum_{i=0}^{r_l} \lambda_{i+1}^2}, \quad \text{for } k = 0, 1, \ldots, r_l.$$

Finally, the ladle method determines the dimension $d$ using the formula

$$\hat{d} = \operatorname*{argmin}_{k=0,\ldots,r_l} \{ f_n(k) + g_n(k) \}.$$

## 4. Theoretical Results

For nonlinear SDR, our primary objective is to identify functions within the central class $\mathfrak{M}_{Y|X}$. The population form of (6) is defined as

$$\begin{aligned} L(f,t) =& \lambda \operatorname{Var}(f(X)) + \lambda t^2 \\ & + E\big[ \rho\big( -\widetilde{Y}\{ f(X) - E(f(X)) - t \} \big) \big], \end{aligned} \tag{7}$$

where $t \in \mathbb{R}$, $f$ belongs to a function class $\mathcal{F}$, and $\rho$ represents any nondecreasing convex loss function. The following theorem shows that it can achieve effective nonlinear SDR.

**Theorem 4.1.** *Suppose that*

*1. $\mathcal{F}$ is a dense subset of $L_2(P_X)$,*

*2. $Y \perp\!\!\!\perp X | \boldsymbol{f_0}(X)$.*

*If $(f^*, t^*)$ minimizes (7) over $\mathcal{F} \times \mathbb{R}$, then $f^*(X) \in \mathfrak{M}_{Y|X}$.*

Linear SDR methods typically rely on the linear conditional mean assumption concerning $E\big( X | \eta^\top X \big)$ to ensure unbiasedness (Li, 1991; Li & Dong, 2009). Remarkably, for the nonlinear method proposed here, such stringent conditions are unnecessary. The first condition of Theorem 4.1 can be satisfied by continuous function class, RKHS with a Gaussian kernel, or $L_2(P_X)$ itself. In general, we usually choose $\mathcal{F} = L_2(P_X)$ directly for population optimization. By Definition 2.1, Theorem 4.1 establishes the validity of our nonlinear SDR method (7) since the optimal solution $f^*$ is unbiased. In other words, we can conclude that $\sigma(f^*(X)) \subseteq \sigma(\boldsymbol{f_0}(X))$. The significance of Theorem 4.1 lies in its transformation of an abstract nonlinear SDR problem into a specific and solvable optimization problem (7). It is essential to emphasize that our goal is to identify functions that capture as many nonlinear relationships between $Y$ and $X$ as possible, rather than approximating specific predetermined functions, such as $\boldsymbol{f_0}$.

More generally, given some function class $\mathcal{F}_n$, below we prove that the sample optimization (6) over $\mathcal{F}_n \times \mathbb{R}$ can converge to the population optimization (7) over $L_2(P_X) \times \mathbb{R}$

at a certain convergence rate. Before stating the results, we need the following assumptions. Let $(f^*, t^*)$ be a minimizer of $L(f,t)$ over $L_2(P_X) \times \mathbb{R}$.

**Assumption 4.2.** There exists an absolute constant $B > 1$ such that $\|f^*\|_\infty \le B$ and $\|f\|_\infty \le B$ for any $f \in \mathcal{F}_n$.

**Assumption 4.3.** $f^*$ is $\beta$-Hölder continuous, *i.e.*, $|f^*(x_1) - f^*(x_2)| \le L_{hld} \|x_1 - x_2\|^\beta$ for any $x_1, x_2 \in [0,1]^p$, where $L_{hld}$ is some positive constant.

**Assumption 4.4.** Nondecreasing convex loss function $\rho : \mathbb{R} \mapsto \mathbb{R}^+$ is Lipschitz continuous with Lipschitz constant $L_{lip}$, *i.e.*, $|\rho(x) - \rho(y)| \le L_{lip} \|x - y\|$ for any $x, y \in \mathbb{R}$.

**Assumption 4.5.** There exist an universal constant $C$ and a parameter $V$ depending on $\mathcal{F}_n$ such that

$$\log N(\epsilon, \mathcal{F}_n, \| \cdot \|_{L_2(Q)}) \le CV \{ 1 + \log(1/\epsilon) \},$$

where $Q$ refers to $P_X$ and $P_n$, the empirical probability measure of $X$.

Assumption 4.2 states that the ground-truth function $f^*$ and neural networks are bounded. The parameter $\beta$ in Assumption 4.3 characterizes the smoothness or regularity of $f^*$. A higher value of $\beta$ indicates greater smoothness, which generally facilitates more accurate approximation. The convergence rate of our estimator is jointly determined by $\beta$ and the input dimension $p$; see Corollary 4.7 for details. Assumption 4.2 and 4.3 are common requirements in many existing works (Györfi et al., 2002; Schmidt-Hieber, 2020; Farrell et al., 2021). Assumption 4.4, crucial for deriving the convergence result, is satisfied by common loss functions such as hinge loss and Huber loss. Regarding the assumption 4.5, this upper bound always holds for any VC class $\mathcal{F}_n$ with the parameter $V$ being its VC dimension (see Theorem 2.6.7 in Vaart & Wellner (1996)), such as splines, RKHS with a polynomial kernel, and deep neural network class. The parameter $V$ is a measure of the size (capacity, complexity, or expressive power) of a hypothesis class $\mathcal{F}_n$, which is relative to the trade-off between statistical error and approximation error of our estimator. Classes with larger $V$ have stronger approximation capabilities but also carry a higher risk of overfitting.

When $\rho(x) = \max\{1 + x, 0\}$, optimizing (6) over RKHS and $\mathbb{R}$ is similar to kernel principal SVM (Li et al., 2011). Flexibility in the optimization range is another advantage over classical SDR methods.

**Theorem 4.6.** *Let $(\hat{f}, \hat{t})$ be the minimizer of $L_n(f,t)$ over $\mathcal{F}_n \times \mathbb{R}$ and $(f^*, t^*)$ be the minimizer of $L(f,t)$ over $L_2(P_X) \times \mathbb{R}$. Let $\delta > 0$. Under the assumptions 4.2, 4.4 and 4.5, if $n \ge V$, the following inequality holds with probability at least $1 - 4\delta$*

$$L(\hat{f}, \hat{t}) - L(f^*, t^*) =$$
$$\mathcal{O}\left( \frac{V}{n} \log \frac{n}{V} + \inf_{f \in \mathcal{F}_n} \|f - f^*\|_\infty + \frac{\log(1/\delta)}{n} \right).$$

Now we focus on the neural network class. By Theorem 2.6.7 in Vaart & Wellner (1996) and Theorem 7 in Bartlett et al. (2019), the log covering number of the ReLU-activated neural network class $\mathcal{F}_n$ with respect to $L_2(Q)$ norm for any probability measure $Q$ can be bounded by

$$\log N\left(\epsilon, \mathcal{F}_n, \|\cdot\|_{L_2(Q)}\right) \leq K_1 \cdot \mathrm{VC}(\mathcal{F}_n)\{1 + \log(1/\epsilon)\}$$
$$\leq K_2 \cdot \mathcal{SL}\log(\mathcal{S})\{1 + \log(1/\epsilon)\},$$

where $K_1, K_2$ are two universal constants, $\mathrm{VC}(\mathcal{F}_n)$ is the VC dimension of the neural network class, $\mathcal{L}$ is the depth of networks, and $\mathcal{S}$ is the total number of network parameters. Hence the neural network class satisfies the assumption 4.5 with $V \leq \mathcal{SL}\log(\mathcal{S})$. Concerning the approximation error, $\beta$-Hölder continuous function $f^*$ can be effectively approximated by ReLU networks. Specifically, considering ReLU neural network class $\mathcal{F}_n$ with width $\mathcal{N} = 3^{p+3}\max\left(p\lfloor N^{1/p}\rfloor, N+1\right)$ and depth $\mathcal{L} = 12L + 14 + 2p$, Theorem 1.1 in Shen (2020) tells that

$$\inf_{f \in \mathcal{F}_n} \|f - f^*\|_{\infty} = \mathcal{O}\left((NL)^{-\frac{2\beta}{p}}\right).$$

Here $N$ and $L$ stand for arbitrary positive constants determining the width $\mathcal{N}$ and the depth $\mathcal{L}$ of neural networks, respectively. With these preparations, we have the following result on the function class of width-fixed ReLU neural networks.

**Corollary 4.7.** *For any arbitrary* $N \in \mathbb{N}^+$, *consider* $\mathcal{F}_n$ *as the scalar-valued ReLU neural network class with width* $\mathcal{N} = 3^{p+3}\max\left(p\lfloor N^{1/p}\rfloor, N+1\right)$ *and depth* $\mathcal{L} = 12n^{\frac{p}{2(p+2\beta)}} + 14 + 2p$. *Let* $\delta > 0$. *Under the assumption 4.2, 4.3 and 4.4, if* $n \geq \mathcal{SL}\log(\mathcal{S})$, *the following result holds with probability at least* $1 - 4\delta$

$$L(\hat{f}, \hat{t}) - L(f^*, t^*) = \mathcal{O}\left(n^{-\frac{2\beta}{p+2\beta}}\{1 + \log(1/\delta)\}\log n\right).$$

*Further,*

$$\|\hat{f} - f^*\|^2_{L_2(P_X)} = \mathcal{O}\left(n^{-\frac{2\beta}{p+2\beta}}\{1 + \log(1/\delta)\}\log n\right).$$

Stone (1982) proved the minimax rate $n^{-\frac{2\beta}{p+2\beta}}$ for nonparametric regression under the Hölder continuous assumption. Remarkably, our nonlinear SDR method, leveraging deep neural networks, achieves this optimal rate up to $\log n$. Theorem 4.1 shows that $f^*$ is a nonlinear SDR function, while Corollary 4.7 demonstrates the convergence of $\hat{f}$ to $f^*$, thereby ensuring the rationality of the first step of nonlinear SDR algorithm process in section 3.1. Regarding the way of aggregating the optimal solutions from all different dividing schemes, the second step of the algorithm process is inspired by the kernel PSVM (Li et al., 2011). As in that work, we do not delve into the theoretical justification behind it here. In particular, we call our nonlinear SDR method optimized over neural networks with $\rho(x) = \max\{1 + x, 0\}$ the deep principal support vector machines (DPSVM).

## 5. Numerical Experiments

We evaluate the finite sample performance of the proposed DPSVM method on synthetic data. For comparison, we consider six other nonlinear dimension reduction methods: kernel principal support vector machine (KPSVM) with Gaussian kernel (Li et al., 2011), generalized sliced inverse regression (GSIR) (Lee et al., 2013), kernel sliced inverse regression (KSIR) (Yeh et al., 2009), deep dimension reduction (DDR) (Huang et al., 2024), deep sufficient representation learning (DSRL) (Zheng et al., 2022) and GMDDNet (Chen et al., 2024). KPSVM and DSRL are implemented based on the algorithm described in the original paper. GSIR and KSIR can be implemented by R package `nsdr` and `MAVE`, respectively. DDR and GMDDNet are implemented using the source code provided by the original authors. More details on the computer configuration and hyperparameters are provided in the Appendix.

Throughout the simulation, we set the training sample size $n = 500$ and the input dimension $p = 10$. We first generate the intermediate response $H$ from the following three models:

$$A : H = 4X_1/(0.1 + X_2^2) + \epsilon;$$
$$B : H = \sin\left((X_1^2 + X_2^2)\pi/10\right) + X_1^2 + \epsilon;$$
$$C : H = \log(0.1 + X_1^2 + X_2^2)\sqrt{X_1^2 + X_2^2} + \epsilon,$$

where the noise $\epsilon$ is independent of $X$ and follows a standard t-distribution with degree of freedom 4. Then we generate a random variable $B \in \{0, 1\}$ by Bernoulli distribution with success probability $pr = 0.1$. Then we acquire the final response $Y = 10H$ if $B = 1$ and $Y = H$ otherwise. This design makes $Y$ an outlier with probability $pr$, increasing the difficulty of nonlinear dimension reduction. Additionally, three different distributional scenarios for the predictor vector $X = (X_1, \ldots, X_p)^{\top}$ are involved:

$$\mathrm{I} : \ X \sim N(\mathbf{0}_p, \mathbf{I}_p),$$
$$\mathrm{II} : \ X \sim t_4(\mathbf{0}_p, \mathbf{I}_p),$$
$$\mathrm{III} : \ X \sim t_4(\mathbf{0}_p, \mathbf{S}_p),$$

where $\mathbf{I}_p$ is the $p \times p$ identity matrix, $\mathbf{S}_p$ is a $p \times p$ matrix with entry $\mathbf{S}_{ij} = 0.5^{|i-j|}$, and $N(\mathbf{A}, \mathbf{B})$ is a multivariate normal distribution with mean $\mathbf{A}$ and covariance $\mathbf{B}$, and $t_k(\mathbf{A}, \mathbf{B})$ is a multivariate t-distribution with mean $\mathbf{A}$, covariance $\mathbf{B}$ and degree of freedom $k$. The true nonlinear SDR functions $f^*$ for the model $A, B, C$ are $4X_1/(0.1 + X_2^2)$, $\sin\left((X_1^2 + X_2^2)\pi/10\right) + X_1^2$ and $\log(0.1 + X_1^2 + X_2^2)\sqrt{X_1^2 + X_2^2}$, respectively. The structure dimension is assumed to be known as 1. We use the empirical distance correlation $\mathrm{DR}_n\left(f^*(X), \hat{f}(X)\right)$ developed in Szkely et al. (2007) to reflect the merit of the estimated dimension reduction function. The detailed formulation of

empirical distance correlation is given in the Appendix. In brief, A higher distance correlation indicates more accuracy in extracting nonlinear information. The average distance correlations (with standard deviations in parentheses) over 100 simulation runs are shown in Table 1. Our method outperforms other methods in seven of the nine settings. This suggests that our approach is more robust to outliers, largely due to the process of bisecting response values (LVR or OVA step in our algorithm).

*Table 1.* Average distance correlations (standard deviations) of different methods for all settings over 100 simulation runs. Boldfaced numbers indicate the best performers.

| Setting | DPSVM | KPSVM | GSIR | KSIR | DDR | DSRL | GMDDNet |
|---------|-------|-------|------|------|-----|------|---------|
| A-I | **0.80(0.01)** | 0.77(0.01) | 0.75(0.01) | 0.75(0.02) | 0.63(0.10) | 0.76(0.00) | 0.76(0.01) |
| A-II | **0.74(0.02)** | 0.69(0.03) | 0.73(0.01) | 0.68(0.04) | 0.57(0.08) | 0.67(0.02) | 0.68(0.00) |
| A-III | **0.77(0.02)** | 0.72(0.03) | 0.72(0.01) | 0.68(0.04) | 0.60(0.09) | 0.70(0.00) | 0.69(0.02) |
| B-I | **0.75(0.05)** | 0.56(0.22) | 0.44(0.06) | 0.15(0.27) | 0.45(0.17) | 0.68(0.17) | 0.57(0.14) |
| B-II | **0.71(0.05)** | 0.59(0.10) | 0.55(0.04) | 0.28(0.18) | 0.55(0.13) | 0.44(0.01) | 0.48(0.17) |
| B-III | 0.75(0.05) | 0.61(0.12) | 0.57(0.04) | 0.27(0.14) | 0.58(0.11) | **0.87(0.02)** | 0.56(0.18) |
| C-I | **0.76(0.04)** | 0.71(0.15) | 0.58(0.05) | 0.20(0.29) | 0.54(0.17) | 0.69(0.19) | 0.79(0.02) |
| C-II | **0.78(0.04)** | 0.73(0.06) | 0.70(0.03) | 0.35(0.11) | 0.62(0.12) | 0.78(0.14) | 0.59(0.15) |
| C-III | 0.81(0.03) | 0.76(0.03) | 0.72(0.02) | 0.35(0.16) | 0.68(0.11) | **0.89(0.07)** | 0.61(0.19) |

**MNIST database.** The MNIST dataset contains $60,000$ training images and $10,000$ testing images of hand-written digits. Here, we take DPSVM and DDR as the representatives of deep methods for 2D visualization. LeNet (Lecun et al., 1998) is used as the basic neural network structure for both DPSVM and DDR. The ladle method in section 3.2 gives the structural dimension estimation $\hat{d} = 9$. Then we train DPSVM and DDR respectively to obtain nine estimated dimension reduction functions. Finally, we train the logistic regression on training images with corresponding nine-dimensional representations of original features and get the classification accuracy result on testing images. The accuracy is $0.9862$ and $0.9883$ for DPSVM and DDR, respectively. As a reference, the accuracy of direct LeNet using the original features is $0.9899$. The closeness of the results suggests that both our method and DDR are effective in extracting nonlinear information from the regression relationship. Lastly, we plot the projections of the original features along the first two estimated dimension reduction directions of DPSVM and DDR to compare visualization effects (see Figure 2). It appears that DPSVM exhibits stronger discriminant power compared to DDR, as the points for each class cluster more distinctly.

**Other datasets.** The datasets[1] involved are *Communities and Crime* (CRIME) for regression tasks; *Connectionist Bench Sonar* (SONAR), *Optical Recognition of Handwritten Digits* (OPTDIGITS), *Breast Cancer Wisconsin* (WDBC) for additional classification tasks. For each dataset, we randomly split the data, using $2/3$ for the training set and $1/3$ for the testing set. Different nonlinear dimension reduction

---

[1] All the datasets are downloaded from UCI machine learning repository

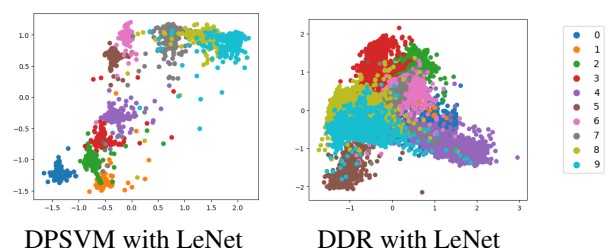

DPSVM with LeNet  DDR with LeNet

*Figure 1.* 2D visualization of MNIST using DPSVM and DDR.

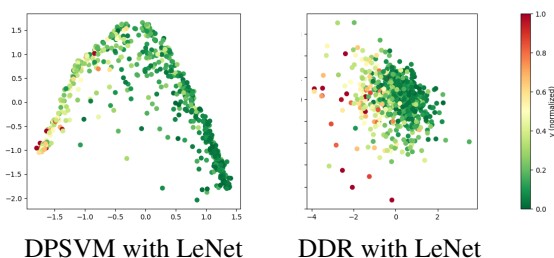

DPSVM with LeNet  DDR with LeNet

*Figure 2.* 2D visualization of CRIME using DPSVM and DDR.

methods are applied to the training set to extract a low-dimensional representation. Then, we fit either linear regression or logistic regression to the representation, depending on whether it is a regression task or a classification task. The mean square error for regression and classification accuracy for classification are calculated on the testing set. To reduce randomness, we repeat the above splitting-extraction-fitting process 100 times. The KSIR method consistently incurs errors on the OPTDIGITS dataset across all 100 repetitions, so its result is excluded from the plot. Note that the y-axis of the first three plots represents accuracy (higher values are better), while the y-axis of the last plot represents the mean square error (lower values are better). Our method shows comparable performance across all datasets, highlighting its competitiveness. Additionally, we present a 2D visualization comparing our approach with the DDR method on the CRIME dataset. The results indicate that our method yields more visually coherent representations.

## 6. Conclusions and Limitations

In this paper, we present a new approach to nonlinear sufficient dimension reduction that avoids the issues associated with generalized eigendecomposition of $n \times n$ matrices in conventional methods. The approach includes kernel principle SVM (Li et al., 2011) and extends it by incorporating neural networks for more flexible deep nonlinear SDR, particularly beneficial for severe data contamination, large samples and complex inputs. The fusion of deep learning with nonlinear SDR brings new energy to the SDR field. From a theoretical standpoint, we provide guarantees of un-

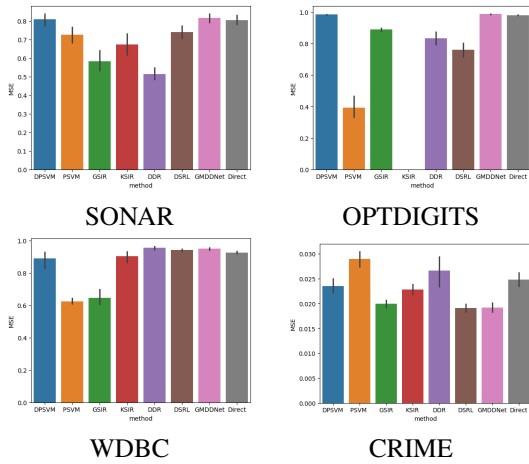

*Figure 3.* Performance of different methods on different datasets

## Impact Statement

This paper presents work whose goal is to advance the field of sufficient dimension reduction. There are many potential societal consequences of our work, none which we feel must be specifically highlighted here.

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

biasedness with respect to the central $\sigma$-field and establish strong convergence rates for estimation error. Numerical simulations and real data analysis demonstrate the exceptional performance of our method.

The limitation of our method is its speed. For real-valued $Y$, we must first discretize $Y$ into binary variables. Typically, we perform this discretization 10 times with different split schemes and fit neural networks on each discretization. Additionally, discretizing a scalar $Y$ is typically based on quantiles, but calculating quantiles for multivariate $Y$ is challenging and requires more effort.

## Acknowledgements

We thank all the anonymous reviewers for their helpful comments and suggestions. Yinfeng Chen is supported by the National Key R&D Program of China (Grant No. 2023YFA1008700 and 2023YFA1008703). Rui Qiu is supported by the Postdoctoral Fellowship Program and China Postdoctoral Science Foundation (Grant No. 2024M760060 and BX20250069). Jin Liu is supported by the National Natural Science Foundation of China (Grant No. 12371283), University Development Fund (Grant No. UDF01003033) from The Chinese University of Hong Kong, Shenzhen, the Program for Guangdong Introducing Innovative and Entrepreneurial Teams (Grant No. 2023ZT10X044), Shenzhen Science and Technology Program (Shenzhen Key Laboratory Grant No. ZDSYS20230626091302006), the Guangdong Provincial Key Laboratory of Mathematical Foundations for Artificial Intelligence (Grant No. 2023B1212010001), Shenzhen Stability Science Program and Shenzhen Fundamental Research Program (General Program) (Grant No. JCYJ20240813113518024).

Lee, K.-Y., Li, B., and Chiaromonte, F. A general theory for nonlinear sufficient dimension reduction: Formulation and estimation. *Annals of Statistics*, 41(1):221 – 249, 2013.

Li, B. and Dong, Y. Dimension reduction for nonelliptically distributed predictors. *Annals of Statistics*, 37(3):1272–1298, 2009.

Li, B. and Wang, S. On directional regression for dimension reduction. *Journal of the American Statistical Association*, 102(479):997–1008, 2007.

Li, B., Artemiou, A., and Li, L. Principal support vector machines for linear and nonlinear sufficient dimension reduction. *Annals of Statistics*, 39(6):3182–3210, 2011.

Li, K.-C. Sliced inverse regression for dimension reduction. *Journal of the American Statistical Association*, 86(414):316–327, 1991.

Loyal, J. D., Zhu, R., Cui, Y., and Zhang, X. Dimension reduction forests: local variable importance using structured random forests. *Journal of Computational and Graphical Statistics*, 31(4):1104–1113, 2022.

Luo, W. and Li, B. Combining eigenvalues and variation of eigenvectors for order determination. *Biometrika*, 103(4):875–887, 2016.

Ma, Y. and Zhu, L. A review on dimension reduction. *International Statistical Review*, 81(1):134–150, 2013.

Massart, P. *Concentration inequalities and model selection*. Springer Verlag, 2003.

Schmidt-Hieber, J. Nonparametric regression using deep neural networks with relu activation function. *Annals of Statistics*, 48(4):18751897, 2020.

Setodji, C. M. and Cook, R. D. K-means inverse regression. *Technometrics*, 46(4):421–429, 2004.

Shen, Z. Deep network approximation characterized by number of neurons. *Communications in Computational Physics*, 28(5):1768–1811, 2020.

Shin, S. J., Wu, Y., Zhang, H. H., and Liu, Y. Principal weighted support vector machines for sufficient dimension reduction in binary classification. *Biometrika*, 104(1):67–81, 2017.

Stone, C. J. Optimal global rates of convergence for nonparametric regression. *Annals of Statistics*, 10(4):1040–1053, 1982.

Szkely, G. J., Rizzo, M. L., and Bakirov, N. K. Measuring and testing dependence by correlation of distances. *Annals of Statistics*, 35(6):2769 – 2794, 2007.

Vaart, A. W. and Wellner, J. A. *Weak convergence and empirical processes*. Springer, 1996.

Wu, H.-M. Kernel sliced inverse regression with applications to classification. *Journal of Computational and Graphical Statistics*, 17(3):590–610, 2008.

Xia, Y., Tong, H., Li, W. K., and Zhu, L.-X. An adaptive estimation of dimension reduction space. *Journal of the Royal Statistical Society: Series B (Statistical Methodology)*, 64(3):363–410, 2002.

Yeh, Y.-R., Huang, S.-Y., and Lee, Y.-J. Nonlinear dimension reduction with kernel sliced inverse regression. *IEEE Transactions on Knowledge and Data Engineering*, 21(11):1590–1603, 2009.

Yin, X., Li, B., and Cook, R. D. Successive direction extraction for estimating the central subspace in a multiple-index regression. *Journal of Multivariate Analysis*, 99(8):1733–1757, 2008.

Zheng, S., Lin, Y., and Huang, J. Deep sufficient representation learning via mutual information, 2022. URL https://arxiv.org/abs/2207.10772.

Zhou, J. and Zhu, L. Principal minimax support vector machine for sufficient dimension reduction with contaminated data. *Computational statistics & data analysis*, 94:33–48, 2016.

# A. Additional materials

## A.1. Illustrative examples

To illustrate the motivation behind sufficient dimension reduction (SDR), we present two simple examples.

### EXAMPLE 1: CLASSIFICATION PROBLEM

Consider the classification problem:
$$Y = \mathbb{1}\left(X_1^2 + X_2^2 - 1 + \varepsilon > 0\right),$$
where $X_1, X_2 \sim \mathcal{N}(0, 1)$ and $\varepsilon \sim \mathcal{N}(0, 0.2)$. Then, the normalized SDR function is given by
$$f_0(X) = \frac{X_1^2 + X_2^2}{2} - 1.$$
Here, a linear transformation is applied to ensure $\mathbb{E}(f_0(X)) = 0, \quad \mathrm{Var}(f_0(X)) = 1$.

### EXAMPLE 2: MULTIVARIATE REGRESSION PROBLEM

Now, consider a multivariate regression problem where $Y \in \mathbb{R}^2$:
$$Y_1 = \sin(X_1 + X_2^2) + \varepsilon_1, \quad Y_2 = \cos(X_3 X_4) + \varepsilon_2,$$
where $X_1, X_2, X_3, X_4 \sim \mathcal{N}(0, 1)$ and $\varepsilon_1, \varepsilon_2 \sim \mathcal{N}(0, 0.2)$. In this case, the normalized SDR function is
$$\boldsymbol{f}_0(X) = \begin{pmatrix} a_0 \sin(X_1 + X_2^2) + b_0 \\ a_1 \cos(X_3 X_4) + b_1 \end{pmatrix},$$
where $a_i$ and $b_i, i = 1, 2$, are introduced to ensure $\mathbb{E}(\boldsymbol{f}_0(X)) = 0, \quad \mathrm{Var}(\boldsymbol{f}_0(X)) = I_2$.

### EXAMPLE 3: GENERAL FORMULATION FOR REGRESSION

More generally, we consider the model
$$Y = \boldsymbol{g}(\boldsymbol{f}_0(X)) + \varepsilon,$$
where $\boldsymbol{f}_0 : \mathbb{R}^p \to \mathbb{R}^d$ and $\boldsymbol{g} : \mathbb{R}^d \to \mathbb{R}^q$. Our goal is to identify the low-dimensional nonlinear representation $\boldsymbol{f}_0(X)$. In this general setting, traditional linear sufficient dimension reduction breaks down.

## A.2. Definition of empirical distance correlation

The empirical distance covariance between two random vectors $Y, Y'$ is defined as
$$\mathrm{DC}_n\left(Y, Y'\right) = \binom{n}{4}^{-1} \sum_{1 \leq \pi_1 \neq \pi_2 \neq \pi_3 \neq \pi_4 \leq n} h\left(\left(Y_{\pi_1}, Y'_{\pi_1}\right), \left(Y_{\pi_2}, Y'_{\pi_2}\right), \left(Y_{\pi_3}, Y'_{\pi_3}\right), \left(Y_{\pi_4}, Y'_{\pi_4}\right)\right),$$
where $\{Y_i\}_{i=1}^n, \{Y'_i\}_{i=1}^n$ are the samples of $Y$ and $Y'$, and
$$\begin{aligned} h\left((Y_1, Y'_1), \ldots, (Y_4, Y'_4)\right) = &\frac{1}{24} \sum_{1 \leq i \neq j \neq k \neq l \leq 4} \left(\|Y_i - Y_j\|_2 \|Y'_i - Y'_j\|_2 \right. \\ &\left. - 2\|Y_i - Y_j\|_2 \|Y'_i - Y'_k\|_2 + \|Y_i - Y_j\|_2 \|Y'_k - Y'_l\|_2\right). \end{aligned}$$
Then the empirical distance correlation is defined as
$$\mathrm{DR}_n\left(Y, Y'\right) = \frac{\mathrm{DC}_n\left(Y, Y'\right)}{\sqrt{\mathrm{DC}_n\left(Y, Y\right)}\sqrt{\mathrm{DC}_n\left(Y', Y'\right)}}. \tag{8}$$

## A.3. Simulations for large $p$

Since neural networks require more data to fit overparameterized models, we set the sample size to $n = 5000$ for $p = 100$. The results, shown in Table 2, indicate that our method remains competitive under this configuration.

*Table 2.* Average distance correlations (standard deviations) of different methods for all settings over 100 simulation runs. Bold-faced numbers indicate the best performers.

| Setting | DPSVM | KPSVM | GSIR | KSIR | DDR | DSRL | GMDDNet |
|---------|-------|-------|------|------|-----|------|---------|
| A-I | **0.77(0.01)** | 0.72(0.01) | 0.74(0.01) | 0.76(0.01) | 0.55(0.05) | 0.67(0.05) | 0.61(0.06) |
| A-II | **0.73(0.01)** | 0.65(0.01) | 0.71(0.01) | 0.68(0.01) | 0.50(0.04) | 0.62(0.05) | 0.62(0.04) |
| A-III | **0.75(0.01)** | 0.65(0.01) | 0.63(0.01) | 0.69(0.01) | 0.51(0.06) | 0.68(0.03) | 0.65(0.03) |
| B-I | **0.70(0.03)** | 0.45(0.03) | 0.06(0.01) | nan(nan) | 0.42(0.05) | 0.69(0.05) | 0.47(0.08) |
| B-II | **0.67(0.03)** | 0.39(0.03) | 0.45(0.03) | 0.25(0.19) | 0.53(0.09) | 0.67(0.04) | 0.64(0.06) |
| B-III | 0.68(0.04) | 0.41(0.02) | 0.45(0.02) | 0.35(0.25) | 0.54(0.11) | **0.69(0.02)** | 0.63(0.06) |
| C-I | **0.71(0.04)** | 0.58(0.02) | 0.08(0.02) | 0.48(0.64) | 0.38(0.14) | 0.55(0.24) | 0.53(0.14) |
| C-II | 0.64(0.16) | 0.54(0.02) | 0.59(0.02) | 0.26(0.05) | 0.63(0.04) | **0.75(0.03)** | 0.70(0.06) |
| C-III | **0.73(0.02)** | 0.59(0.01) | 0.55(0.01) | 0.25(0.05) | 0.61(0.09) | 0.73(0.03) | 0.71(0.02) |

## A.4. Analysis of the computational complexity

Let $n, p, \mathcal{L}$, and $\mathcal{N}$ represent the sample size, input dimension, number of hidden layers, and maximum width of the hidden layers, respectively. Let the batch size and training epochs be $b$ and $t$, respectively. The computation overhead of each iteration is primarily attributed to calculating the gradient of that batch. This computation overhead is $\mathcal{O}(b\mathcal{L}\max\{p, \mathcal{N}\}^2)$. Considering that we need to update $n/b$ batches for each epoch, the total complexity of each binarization is

$$\mathcal{O}(nt\mathcal{L}\max\{p, \mathcal{N}\}^2).$$

Additionally, using the LVR procedure involves $h$ binarizations. After performing $h$ binarizations, we need to construct the $h \times h$ matrix based on the inner product of two $h \times n$ matrices and compute its eigendecomposition. These two operations require $\mathcal{O}(h^2 n)$ and $\mathcal{O}(h^3)$ computations, respectively. Therefore, the final computational cost is

$$\mathcal{O}(hnt\mathcal{L}\max\{p, \mathcal{N}\}^2 + h^2 n + h^3).$$

In contrast, deep methods without binarization incur a slightly better computational cost,

$$\mathcal{O}(nt\mathcal{L}\max\{p, \mathcal{N}\}^2).$$

The primary computational complexity of GSIR/PSVM stems from the inversion of an $n \times n$ matrix, which is $\mathcal{O}(n^3)$ for the naive inversion algorithm. When comparing this with the complexity $\mathcal{O}(hnt\mathcal{L}\max\{p, \mathcal{N}\}^2)$, it is evident that the computational advantage of neural networks becomes apparent only when $n$ exceeds a certain threshold. Otherwise, the computational overhead of gradient evaluation becomes more expensive.

The above conclusion is validated by empirical evaluation in Figure 4. Here, deepsvm represents our proposed method, psvm corresponds to KPSVM, and deepsdr refers to DDR. The other names correspond to their respective original methods as indicated.

## A.5. Computer configuration and network architecture

All the experiments are run on a computer with 80 Intel(R) Xeon(R) Gold 5218R CPU @ 2.10GHz CPU and 251 GB memory. The batch size and epoch number are set as 100 and the optimizer is set as adam with default parameters. This network structure follows an expansion-contraction pattern, a reverse pattern of the bottleneck structure in auto-encoders. The initial increase in width is to map the original features to a higher-dimensional space, similar to the kernel trick used in a reproducing kernel Hilbert space. The subsequent decreasing width is employed to perform dimension reduction, ultimately obtaining a low-dimensional embedding. The default network is a feedforward ReLU network with hidden dimensions

$$2^{D_1}, 2^{D_1+1}, 2^{D_1}, 2^{D_1-1}, \ldots, 16, 1,$$

where $D_1 = \lfloor \log_2 p \rfloor + 1$ and $p$ is the input dimension. Our method performs well with a simple feedforward network, so we did not explore more complex architectures. Specifically, we generate a default neural network with the following Python code.

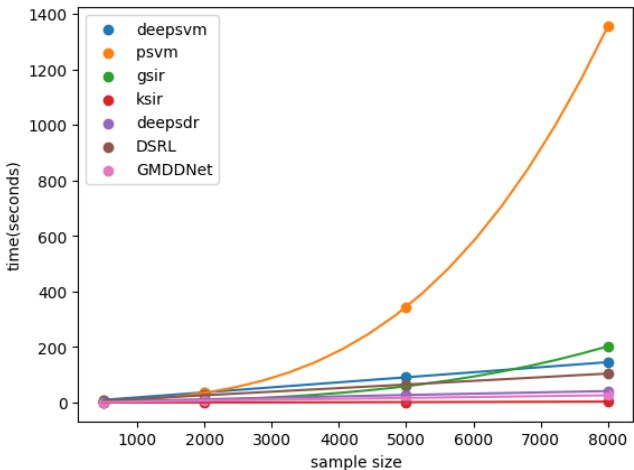

*Figure 4.* Computational time comparison of different methods.

```python
def generate_default_net(input_dim, first_hidden_width=None, last_hidden_width=16,
                                        increase_count=2,  output_dim=1):
    if first_hidden_width is None:
        first_hidden_width = 2 ** (int(np.log2(input_dim)) + 1)
    net = nn.Sequential()
    net.append(nn.Linear(input_dim, first_hidden_width))
    net.append(nn.ReLU())
    next_width = first_hidden_width
    for i in range(increase_count - 1):
        net.append(nn.Linear(next_width, next_width * 2))
        net.append(nn.ReLU())
        next_width = next_width * 2
    for i in range(increase_count - 1):
        net.append(nn.Linear(next_width, next_width // 2))
        net.append(nn.ReLU())
        next_width = next_width // 2
    if last_hidden_width is not None:
        while(next_width // 2 >= last_hidden_width):
            net.append(nn.Linear(next_width, next_width // 2))
            net.append(nn.ReLU())
            next_width = next_width // 2
    net.append(nn.Linear(next_width, output_dim))
    return net
```

## A.6. Sensitivity analysis with different hyperparameters

To evaluate the robustness of our method, we conduct a sensitivity analysis on various hyperparameters: binarization count, $\lambda$, learning rate, and neural network structure. The results are presented in Table 3. DPSVM is not sensitive to the number of binarizations. The learning rate should be chosen suitably small to ensure optimal performance. As shown in Table 3, smaller values of $\lambda$ lead to improved performance, as it shifts the focus toward dimension reduction rather than variance constraints. Specifically, $\lambda$ should range from 0.01 to 0.1. Different configurations of hidden layers and widths of neural networks have the most significant impact on performance in settings B and C. Four different structures I–IV are generated by the code

```python
generate_default_net(10, last_hidden_width=16, increase_count=2) # default
generate_default_net(10, last_hidden_width=8, increase_count=2)
generate_default_net(10, last_hidden_width=16, increase_count=3)
generate_default_net(10, first_hidden_width=32, last_hidden_width=32, increase_count=2)
```

The optimal configuration uses the default network (Setting I), 10 binarizations to balance performance and computational cost, $\lambda = 0.01$, and a learning rate of $0.001$, chosen by cross-validation.

| Setting | 5 | 10 | 15 | 20 |
|---|---|---|---|---|
| A-I | 0.80(0.01) | 0.81(0.01) | 0.81(0.01) | 0.82(0.02) |
| A-II | 0.74(0.01) | 0.73(0.02) | 0.73(0.01) | 0.74(0.02) |
| A-III | 0.76(0.01) | 0.77(0.01) | 0.77(0.01) | 0.77(0.02) |
| B-I | 0.73(0.04) | 0.73(0.05) | 0.74(0.04) | 0.73(0.04) |
| B-II | 0.71(0.04) | 0.70(0.03) | 0.72(0.04) | 0.70(0.05) |
| B-III | 0.74(0.05) | 0.75(0.06) | 0.76(0.05) | 0.75(0.05) |
| C-I | 0.75(0.04) | 0.76(0.02) | 0.79(0.03) | 0.78(0.03) |
| C-II | 0.77(0.02) | 0.77(0.03) | 0.77(0.05) | 0.78(0.04) |
| C-III | 0.80(0.03) | 0.82(0.03) | 0.81(0.03) | 0.81(0.02) |

Sensitivity analysis with a different number of binarizations.

| Setting | 0.01 | 0.1 | 1 | 10 |
|---|---|---|---|---|
| A-I | 0.79(0.02) | 0.81(0.01) | 0.78(0.02) | 0.46(0.08) |
| A-II | 0.72(0.02) | 0.75(0.02) | 0.70(0.02) | 0.32(0.08) |
| A-III | 0.75(0.03) | 0.76(0.02) | 0.72(0.02) | 0.34(0.09) |
| B-I | 0.76(0.03) | 0.74(0.05) | 0.68(0.07) | 0.31(0.06) |
| B-II | 0.76(0.03) | 0.71(0.04) | 0.61(0.08) | 0.27(0.07) |
| B-III | 0.79(0.03) | 0.75(0.03) | 0.68(0.05) | 0.34(0.07) |
| C-I | 0.77(0.07) | 0.76(0.04) | 0.71(0.03) | 0.36(0.08) |
| C-II | 0.82(0.02) | 0.78(0.04) | 0.72(0.04) | 0.36(0.07) |
| C-III | 0.84(0.03) | 0.81(0.03) | 0.76(0.04) | 0.39(0.08) |

Sensitivity analysis with a different $\lambda$.

| Setting | 0.001 | 0.01 | 0.1 | 1 |
|---|---|---|---|---|
| A-I | 0.81(0.01) | 0.82(0.01) | 0.64(0.12) | 0.09(0.05) |
| A-II | 0.74(0.02) | 0.78(0.01) | 0.59(0.11) | 0.07(0.03) |
| A-III | 0.76(0.03) | 0.80(0.01) | 0.61(0.14) | 0.09(0.06) |
| B-I | 0.75(0.06) | 0.68(0.04) | 0.56(0.16) | 0.07(0.02) |
| B-II | 0.72(0.05) | 0.65(0.04) | 0.60(0.08) | 0.08(0.04) |
| B-III | 0.74(0.04) | 0.67(0.04) | 0.60(0.13) | 0.11(0.08) |
| C-I | 0.77(0.03) | 0.69(0.04) | 0.51(0.21) | 0.09(0.06) |
| C-II | 0.78(0.03) | 0.74(0.05) | 0.60(0.15) | 0.09(0.04) |
| C-III | 0.80(0.03) | 0.77(0.04) | 0.61(0.17) | 0.07(0.04) |

Sensitivity analysis with a different learning rate.

| Setting | I | II | III | IV |
|---|---|---|---|---|
| A-I | 0.79(0.01) | 0.80(0.02) | 0.81(0.02) | 0.80(0.03) |
| A-II | 0.74(0.02) | 0.74(0.03) | 0.75(0.01) | 0.74(0.02) |
| A-III | 0.78(0.01) | 0.77(0.02) | 0.79(0.02) | 0.76(0.03) |
| B-I | 0.74(0.03) | 0.54(0.04) | 0.59(0.05) | 0.57(0.05) |
| B-II | 0.72(0.03) | 0.54(0.05) | 0.52(0.05) | 0.54(0.05) |
| B-III | 0.77(0.02) | 0.63(0.03) | 0.58(0.03) | 0.60(0.03) |
| C-I | 0.75(0.04) | 0.62(0.06) | 0.63(0.04) | 0.60(0.06) |
| C-II | 0.77(0.04) | 0.66(0.05) | 0.63(0.04) | 0.64(0.04) |
| C-III | 0.82(0.02) | 0.69(0.02) | 0.64(0.03) | 0.67(0.04) |

Sensitivity analysis with a different network.

*Table 3.* Results of sensitivity analysis with different hyperparameters

# B. Proofs

**Lemma B.1.** *Under the assumption 4.4, for each $t \in \mathbb{R}$, the function $f \mapsto L(f,t)$ is continuous with respect to the $L_2(P_X)$ norm.*

*Proof.* The proof follows the same idea as Lemma 1 in Li et al. (2011). For any $f_1, f_2 \in L_2(P_X)$, the following inequality holds

$$\left| L(f_1,t) - L(f_2,t) \right| \leq \lambda \left| \operatorname{Var}(f_1(X)) - \operatorname{Var}(f_2(X)) \right|$$
$$+ E \left| \rho\left(-\widetilde{Y}\{f_1(X) - E(f_1(X)) - t\}\right) - \rho\left(-\widetilde{Y}\{f_2(X) - E(f_2(X)) - t\}\right) \right|.$$

By Cauchy-Schwarz inequality,

$$\left| \operatorname{Var}(f_1(X)) - \operatorname{Var}(f_2(X)) \right|$$
$$= \left| \operatorname{Var}(f_1(X) - f_2(X) + f_2(X)) - \operatorname{Var}(f_2(X)) \right|$$
$$\leq \left| \operatorname{Var}(f_1(X) - f_2(X)) \right| + 2 \left| \operatorname{Var}(f_1(X) - f_2(X)) \operatorname{Var}(f_2(X)) \right|^{1/2}$$
$$\leq \|f_1(X) - f_2(X)\|_{L_2(P_X)}^2 + 2\|f_1(X) - f_2(X)\|_{L_2(P_X)} \|f_2(X)\|_{L_2(P_X)}.$$

By Lipschitz continuity of $\rho$,

$$E \left| \rho\left(-\widetilde{Y}\{f_1(X) - E(f_1(X)) - t\}\right) - \rho\left(-\widetilde{Y}\{f_2(X) - E(f_2(X)) - t\}\right) \right|$$
$$\leq L_{lip} \cdot E \left| \widetilde{Y}\{f_1(X) - E(f_1(X)) - f_2(X) + E(f_2(X))\} \right|$$
$$\leq 2L_{lip} \cdot E \left| f_1(X) - f_2(X) \right|$$

$$\leq 2L_{lip} \cdot \|f_1(X) - f_2(X)\|_{L_2(P_X)}.$$

Then we can conclude that the function $f \mapsto L(f, t)$ is continuous with respect to the $L_2(P_X)$ norm. $\qquad \square$

**Proof of Theorem 4.1.** Without loss of generality, we assume $E(f(X)) = 0$. Note that

$$E\rho(-\widetilde{Y}\{f(X) - t\}) = E\Big(E\Big[\rho(-\widetilde{Y}\{f(X) - t\})|Y, \boldsymbol{f}_0(X)\Big]\Big).$$

By Jensen inequality and conditional independence $Y \perp\!\!\!\perp X | \boldsymbol{f}_0(X)$,

$$E\Big(E\Big[\rho(-\widetilde{Y}\{f(X) - t\})|Y, \boldsymbol{f}_0(X)\Big]\Big) \geq E\Big(\rho\Big(E\Big[-\widetilde{Y}\{f(X) - t\}|Y, \boldsymbol{f}_0(X)\Big]\Big)\Big)$$
$$= E\Big(\rho\Big(-\widetilde{Y}\{E[f(X)|\boldsymbol{f}_0(X)] - t\}\Big)\Big). \tag{9}$$

Now apply the law of total variance, then we get

$$\mathrm{Var}(f(X)) = \mathrm{Var}\big[E\big(f(X)|\boldsymbol{f}_0(X)\big)\big] + E\big[\mathrm{Var}\big(f(X)|\boldsymbol{f}_0(X)\big)\big] \geq \mathrm{Var}\big[E\big(f(X)|\boldsymbol{f}_0(X)\big)\big]. \tag{10}$$

Combining (9) and (10), we have

$$L(f, t) \geq \lambda\,\mathrm{Var}\big[E(f(X)|\boldsymbol{f}_0(X))\big] + \lambda t^2 + E\Big[\rho\Big(-\widetilde{Y}\{E[f(X)|\boldsymbol{f}_0(X)] - t\}\Big)\Big]$$
$$= L\Big(E\big(f(X)|\boldsymbol{f}_0(X)\big), t\Big).$$

The equality in (10) holds when

$$E\big[\mathrm{Var}\big(f(X)|\boldsymbol{f}_0(X)\big)\big] = 0 \quad \text{a.s.}$$

which holds if and only if there is a version of $f$ that is measurable with respect to $\sigma\{\boldsymbol{f}_0(X)\}$. Therefore, if $E\big[\mathrm{Var}\big(f(X)|\boldsymbol{f}_0(X)\big)\big] > 0$ or $f$ is not measurable with respect to $\sigma\{\boldsymbol{f}_0(X)\}$, then

$$L(f, t) > L(g, t)$$

where $g(X) = E\left[f(X)|\boldsymbol{f}_0(X)\right]$ is a function in $L_2(P_X)$. And $\mathcal{F}$ is dense in $L_2(P_X)$, so we can find a function $g_f$ such that

$$\|g_f - g\|_{L_2(P_X)} \leq \epsilon$$

for all $\epsilon > 0$. By Lemma B.1 and the arbitrariness of $\epsilon$, we find $g_f \in \mathcal{F}$ such that

$$L(f, t) > L(g_f, t),$$

which concludes that any function that is not measurable with respect to $\sigma\{\boldsymbol{f}_0(X)\}$ can't be the minimizer.

**Proof of Theorem 4.6.** To facilitate theoretical analysis, we define an intermediate quantity

$$\tilde{L}_n(f, t) = \lambda\,\mathrm{Var}_n(f(X)) + \lambda t^2 + \frac{1}{n}\sum_{i=1}^{n}\rho\big(-\widetilde{Y}_i\{f(X_i) - E(f(X)) - t\}\big) \tag{11}$$
$$= \frac{2\lambda}{n(n-1)}\sum_{1 \leq i < j \leq n}\frac{\{f(X_i) - f(X_j)\}^2 + 2t^2}{2}$$
$$+ \frac{1}{n}\sum_{i=1}^{n}\rho\big(-\widetilde{Y}_i\{f(X_i) - Ef(X) - t\}\big)$$
$$= \frac{2}{n(n-1)}\sum_{1 \leq i < j \leq n}h_{f,t}(Z_i, Z_j),$$

where $h_{f,t}(Z_i, Z_j) = \lambda t^2 + \frac{\lambda \{f(X_i) - f(X_j)\}^2}{2} + \frac{\rho(-\widetilde{Y}_i \{f(X_i) - E(f(X)) - t\}) + \rho(-\widetilde{Y}_j \{f(X_j) - E(f(X)) - t\})}{2}$ is a symmetric kernel and $Z_i = (X_i, \widetilde{Y}_i)$.

Define $q_{f,t}(Z_i, Z_j) = h_{f,t}(Z_i, Z_j) - h_{f^*, t^*}(Z_i, Z_j)$, where $(f^*, t^*)$ is the optimal solution of $L(f, t)$ over $L_2(P_X) \times \mathbb{R}$. Now we consider

$$\tilde{\Lambda}_n(f, t) = \tilde{L}_n(f, t) - \tilde{L}_n(f^*, t^*) = \frac{2}{n(n-1)} \sum_{1 \le i < j \le n} q_{f,t}(Z_i, Z_j).$$

It's obvious that the expectation of $\tilde{\Lambda}_n(f, t)$ is the excess risk $\Lambda(f, t) = L(f, t) - L(f^*, t^*)$.

Given the conditional kernel

$$q_{0,f,t} = \int q_{f,t}(z_1, z_2) dP(z_1) dP(z_2) = \Lambda(f, t),$$

$$q_{1,f,t}(z_1) = \int q_{f,t}(z_1, z_2) dP(z_2),$$

$$q_{2,f,t}(z_1, z_2) = q_{f,t}(z_1, z_2),$$

then the Hoeffding decomposition of $\tilde{\Lambda}_n(f, t)$ is

$$\begin{aligned}
\tilde{\Lambda}_n(f, t) = & q_{0,f,t} + \frac{2}{n} \sum_{i=1}^{n} \{q_{1,f,t}(Z_i) - q_{0,f,t}\} \\
& + \frac{1}{n(n-1)} \sum_{1 \le i \ne j \le n} \{q_2(Z_i, Z_j) - q_{1,f,t}(Z_i) - q_{1,f,t}(Z_j) + q_{0,f,t}\} \\
:= & A_{1,n}(f, t) + A_{2,n}(f, t)
\end{aligned} \tag{12}$$

where $A_{1,n}(f, t) = q_{0,f,t} + \frac{2}{n} \sum_{i=1}^{n} \{q_{1,f,t}(Z_i) - q_{0,f,t}\} = \frac{2}{n} \sum_{i=1}^{n} q_{1,f,t}(Z_i) - \Lambda(f, t)$ and $A_{2,n}(f, t)$ is the third term. Noting that $A_{1,n}(f, t)$ is an independent sum, let a new loss function

$$\ell(f, t, Z) = 2q_{1,f,t}(Z) - \Lambda(f, t).$$

Now we consider a new objective function

$$v_n(f, t) = A_{1,n}(f, t) = \frac{1}{n} \sum_{i=1}^{n} \ell(f, t, Z_i)$$

whose minimizer over $\mathcal{F}_n \times \mathbb{R}$ is denoted by $(f_\ell, t_\ell)$ and centered empirical form is

$$\bar{v}_n(f, t) = \frac{1}{n} \sum_{i=1}^{n} \ell(f, t, Z_i) - \Lambda(f, t).$$

**Step 1: Loss gap.**

The sample estimator $(\hat{f}, \hat{t})$ is obtained by minimizing $L_n$, while our convergence analysis turns to the independent sum $v_n(f, t)$. We should derive the loss gap between $v_n(\hat{f}, \hat{t})$ and $v_n(f_\ell, t_\ell)$ at first. Let the minimizer of $\tilde{L}_n$ be $(\tilde{f}, \tilde{t})$, then

$$\begin{aligned}
0 \le \tilde{L}_n(\hat{f}, \hat{t}) - \tilde{L}_n(\tilde{f}, \tilde{t}) = & (\tilde{L}_n(\hat{f}, \hat{t}) - L_n(\hat{f}, \hat{t})) + (L_n(\hat{f}, \hat{t}) - \tilde{L}_n(\tilde{f}, \tilde{t})) \\
:= & D_1 + D_2.
\end{aligned}$$

The term $D_2$ can be bounded by

$$\begin{aligned}
D_2 = & L_n(\hat{f}, \hat{t}) - L_n(\tilde{f}, \tilde{t}) + L_n(\tilde{f}, \tilde{t}) - \tilde{L}_n(\tilde{f}, \tilde{t}) \\
\le & L_n(\tilde{f}, \tilde{t}) - \tilde{L}_n(\tilde{f}, \tilde{t})
\end{aligned}$$

$$\leq \left| L_n(\tilde{f}, \tilde{t}) - \tilde{L}_n(\tilde{f}, \tilde{t}) \right|$$

$$\leq E_n \left| \rho\big(-\widetilde{Y}\{\tilde{f}(X) - E_n(\tilde{f}(X)) - t\}\big) - \rho\big(-\widetilde{Y}\{\tilde{f}(X) - E(\tilde{f}(X)) - t\}\big) \right|$$

$$\leq L_{lip} \cdot \left| E_n(\tilde{f}(X)) - E(\tilde{f}(X)) \right|.$$

Noting that $\tilde{f}$ is bounded by $B$, by Hoeffding's inequality, with probability at least $1 - \exp(-\frac{2n\delta^2}{B^2})$ it holds that

$$L_{lip} \cdot \left| E_n(\tilde{f}(X)) - E(\tilde{f}(X)) \right| \leq L_{lip}\delta,$$

Or equivalently,

$$D_2 \leq L_{lip}B\sqrt{\frac{\log(1/\delta)}{2n}}$$

with probability at least $1 - \delta$. And $D_1$ can be bounded by the same logic. Therefore, with probability at least $1 - 2\delta$,

$$\left| \tilde{L}_n(\hat{f}, \hat{t}) - \tilde{L}_n(\tilde{f}, \tilde{t}) \right| \leq 2L_{lip}B\sqrt{\frac{\log(1/\delta)}{2n}}.$$

Therefore with probability at least $1 - 2\delta$, we have

$$\tilde{L}_n(\hat{f}, \hat{t}) - \tilde{L}_n(f^*, t^*) - 2L_{lip}B\sqrt{\frac{\log(1/\delta)}{2n}} \leq \tilde{L}_n(\tilde{f}, \tilde{t}) - \tilde{L}_n(f^*, t^*),$$

which implies

$$\tilde{\Lambda}_n(\hat{f}, \hat{t}) - 2L_{lip}B\sqrt{\frac{\log(1/\delta)}{2n}} \leq \tilde{\Lambda}_n(\tilde{f}, \tilde{t}) \leq \tilde{\Lambda}_n(f_\ell, t_\ell)$$

since $(\tilde{f}, \tilde{t})$ is also the minimizer of $\tilde{\Lambda}_n(f, t)$. With $\tilde{\Lambda}_n(f, t) = A_{1,n}(f, t) + A_{2,n}(f, t)$, we further have

$$A_{1,n}(\hat{f}, \hat{t}) - A_{1,n}(f_\ell, t_\ell) \leq 2L_{lip}B\sqrt{\frac{\log(1/\delta)}{2n}} + A_{2,n}(f_\ell, t_\ell) - A_{2,n}(\hat{f}, \hat{t}). \tag{13}$$

Additionally, by corollary 6 in Clémençon et al. (2008), with probability at least $1 - \delta$ it holds that

$$\sup_{f \in \mathcal{F}_n} \left| A_{2,n}(f) \right| \leq C\big(V/n + (\log 1/\delta)/n\big), \tag{14}$$

where $C$ is a constant and $V$ is defined in Assumption 4.5.

Combining (13) and (14), it holds that with probability at least $1 - 3\delta$

$$A_{1,n}(\hat{f}, \hat{t}) - A_{1,n}(f_\ell, t_\ell) \leq 2L_{lip}B\sqrt{\frac{\log(1/\delta)}{2n}} + 2C\big(V/n + (\log 1/\delta)/n\big).$$

Choosing $\delta$ small enough so that $\frac{\log(1/\delta)}{2n} \geq 1$, then we can conclude that with probability at least $1 - 3\delta$,

$$v_n(\hat{f}, \hat{t}) - v_n(f_\ell, t_\ell) \leq \mathcal{O}(V/n + (\log 1/\delta)/n).$$

**Step 2: Pseudo-metric.**

Define the pseudo-metric as $d_p((f_1, t_1), (f_2, t_2)) = C_1(\|f_1 - f_2\|_{L^2(P_X)} + |t_1 - t_2|)$, where the universal constant $C_1$ will be determined later to satisfy some properties. Since

$$q_{1,f,t}(Z) = \lambda t^2 + \frac{\lambda\{f(X) - E(f(X))\}^2}{2} + \frac{\rho\big(-\widetilde{Y}\{f(X) - E(f(X)) - t\}\big)}{2}$$

$$+ \frac{E\rho\left(-\widetilde{Y}\{f(X) - E(f(X)) - t\}\right)}{2},$$

we have

$$
\begin{aligned}
&\text{Var}\left[\ell(f_1, t_1, Z) - \ell(f_2, t_2, Z)\right] \\
&= \text{Var}\left[2q_{1,f_1,t_1}(Z) - 2q_{1,f_2,t_2}(Z)\right] \\
&= \text{Var}\Big[2\lambda\{f_1(X) - E(f_1(X))\}^2 + 2\lambda t_1^2 + \rho(-\widetilde{Y}\{f_1(X) - E(f_1(X)) - t_1\}) \\
&\quad\quad - 2\lambda\{f_2(X) - E(f_2(X))\}^2 - 2\lambda t_2^2 - \rho(-\widetilde{Y}\{f_2(X) - E(f_2(X)) - t_2\})\Big].
\end{aligned}
$$

Noting that $f_1, f_2$ is bounded by $B$, $\text{Var}\left[\lambda\{f_1(X) - E(f_1(X))\}^2 - \lambda\{f_2(X) - E(f_2(X))\}^2\right]$ can be bounded by

$$
\begin{aligned}
&\lambda^2 E\left[\{f_1(X) - E(f_1(X)) + f_2(X) - E(f_2(X))\}^2\{f_1(X) - E(f_1(X)) - f_2(X) + E(f_2(X))\}^2\right] \\
&\leq 32B^2\lambda^2\left[E\{f_1(X) - f_2(X)\}^2 + \{E(f_1(X)) - E(f_2(X))\}^2\right] \\
&\leq 64B^2\lambda^2\|f_1 - f_2\|_{L_2(P_X)}^2.
\end{aligned}
$$

With Lipschitz continuity of $\rho$, the remaining term can be bounded similarly by

$$2L_{lip}^2(\|f_1 - f_2\|_{L^2(P_X)}^2 + |t_1 - t_2|^2)$$

Let $C_1 = 2L_{lip}^2 + 64B^2\lambda^2$. Then

$$\text{Var}\left[\ell(f_1, t_1, Z) - \ell(f_2, t_2, Z)\right] \leq d_p^2((f_1, t_1), (f_2, t_2)).$$

The following deduction is still valid if the zero mean assumption doesn't hold. We can just replace $f$ and $f^*$ by $\bar{f} = f - Ef$ and $\bar{f}^* = f^* - Ef^*$, respectively. With loss of generality, we assume $E(f^*(X)) = 0$ here. Let $f$ be any function such that $E(f(X)) = 0$ and $t$ be any constant, then

$$
\begin{aligned}
&L(f, t) - L(f^*, t^*) \\
&= \lambda\{\text{Var}(f(X)) + t^2\} - \lambda\{\text{Var}(f^*(X)) + t^{*2}\} + E\rho(-\widetilde{Y}\{f(X) - t\}) - E\rho(-\widetilde{Y}\{f^*(X) - t^*\}) \\
&= \lambda E\{f(X) - f^*(X)\}^2 + \lambda|t - t^*|^2 + \lambda E[2\{f(X) - f^*(X)\}f^*(X)] + 2t^*(t - t^*) \\
&\quad + E\rho(-\widetilde{Y}\{f(X) - t\}) - E\rho(-\widetilde{Y}\{f^*(X) - t^*\}) \\
&= \lambda E\{f(X) - f^*(X)\}^2 + \lambda|t - t^*|^2 + A \\
&\geq \lambda E\{f(X) - f^*(X)\}^2 + \lambda|t - t^*|^2 \\
&= \lambda\|f - f^*\|_{L_2(P_X)}^2 + \lambda|t - t^*|^2,
\end{aligned}
$$

if

$$
\begin{aligned}
A &= \lambda E[2\{f(X) - f^*(X)\}f^*(X)] + 2t^*(t - t^*) + E\rho(-\widetilde{Y}\{f(X) - t\}) - E\rho(-\widetilde{Y}\{f^*(X) - t^*\}) \\
&\geq 0.
\end{aligned}
\tag{15}
$$

Now we prove the condition (15) by contradiction. Let's assume $A < 0$. By convexity of loss function $\rho$, the following holds for any $0 \leq c \leq 1$,

$$
\begin{aligned}
&L(cf + (1-c)f^*, ct + (1-c)t^*) - L(f^*, t^*) \\
&\leq \lambda c^2\left[E\{f(X) - f^*(X)\}^2 + |t - t^*|^2\right] + \lambda c E[2\{f(X) - f^*(X)\}f^*(X)] + 2\lambda c t^*(t - t^*) \\
&\quad + cE\rho(-\widetilde{Y}\{f(X) - t\}) - cE\rho(-\widetilde{Y}\{f^*(X) - t^*\}) \\
&= \lambda c^2(E\{f(X) - f^*(X)\}^2 + |t - t^*|^2) + cA.
\end{aligned}
$$

The root of $\lambda c^2 (E\{f(X) - f^*(X)\}^2 + |t - t^*|^2) + cA$ is

$$0 \text{ or } \frac{-2A}{2\lambda\big[E\{f(X) - f^*(X)\}^2 + |t - t^*|^2\big]} > 0.$$

Therefore for some small $0 < c < 1$, it holds that

$$L(cf + (1-c)f^*, ct + (1-c)t^*) < L(f^*, t^*),$$

which contradicts with the fact that $(f^*, t^*)$ is the minimizer.

Therefore, we get the lower bound of $L(f, t) - L(f^*, t^*)$

$$d_p((f_1, t_1), (f_2, t_2)) \le \frac{C_1}{\lambda}\sqrt{L(f, t) - L(f^*, t^*)} := w(\sqrt{L(f, t) - L(f^*, t^*)}),$$

which implies that $w(x) = \frac{C_1}{\lambda}x$ is a linear function of $x$.

**Step 3: $\phi$ bound**

Next we need to find a function $\phi : [0, \infty) \mapsto [0, \infty)$ with the property

$$\sqrt{n}W(\sigma) = \sqrt{n}E\left[\sup_{f \in \mathcal{F}_n, t \in \mathbb{R}, d_p^2((f,t),(f_n^*,t^*)) \le \sigma^2} v_n(f, t) - v_n(f_n^*, t^*)\right] \le \phi(\sigma),$$

where $f_n^* = \operatorname{argmin}_{f \in \mathcal{F}_n} L(f, t)$. Applying Lemma 6.10 in Blanchard et al. (2008), we get

$$
\begin{aligned}
&\sqrt{n}W(\sigma)\\
&\le C\Big(\int_0^\sigma \sqrt{\log N(\delta, \{\ell(f,t) - \ell(f_n^*, t^*) : f \in \mathcal{F}_n, t \in \mathbb{R}, d_p^2((f_n^*, t^*), (f, t)) \le \sigma^2\}, \|\cdot\|_\infty)}d\delta\\
&\quad + \frac{\log N(\sigma, \{\ell(f, t) - \ell(f_n^*, t^*) : f \in \mathcal{F}_n, t \in \mathbb{R}, d_p^2((f_n^*, t^*), (f, t)) \le \sigma^2\}, \|\cdot\|_\infty)}{\sqrt{n}}\Big)\\
&\le C\Big(\int_0^\sigma \sqrt{\log N(\delta, \{f - f_n^* + t - t^* : f \in \mathcal{F}_n, t \in \mathbb{R}, d_p^2((f_n^*, t^*), (f, t)) \le \sigma^2\}, \|\cdot\|_\infty)}d\delta\\
&\quad + \frac{\log N(\sigma, \{f - f_n^* + t - t^* : f \in \mathcal{F}_n, t \in \mathbb{R}, d_p^2((f_n^*, t^*), (f, t)) \le \sigma^2\}, \|\cdot\|_\infty)}{\sqrt{n}}\Big)\\
&\quad (\text{ By Lipschitz continuous condition})\\
&\le C\Big(\int_0^\sigma \sqrt{\log(B/\delta) + H_\infty(\delta)}d\delta + \frac{\log(B/\delta) + H_\infty(\sigma)}{\sqrt{n}}\Big)\\
&\le C\Big(\xi(\sigma) + \frac{(\xi(\sigma))^2}{\sigma^2\sqrt{n}}\Big)(\text{ By } \xi(x) \ge x\sqrt{H_\infty(x)})(\text{ assume } V > B)\\
&:= \phi(\sigma),
\end{aligned}
$$

where $H_\infty(\delta) = \log N(\delta, \mathcal{F}_n, \|\cdot\|_\infty)$ and $\xi(x) = \int_0^x \sqrt{H_\infty(\delta)}d\delta \le C(\sqrt{V}x + \sqrt{V}\int_0^x \log(1/\delta)d\delta)$.

Denote $x_*$ the solution of equation $\xi(x) = \sqrt{n}x^2$ and $\sigma_* = cx_*$ for a suitable choice of constant $c \ge 2C > 2$, Since $x^{-1}\xi(x)$ is a decreasing function, it holds that

$$\xi(\sigma_*) = \xi(cx_*) \le c\xi(x_*) = c\sqrt{n}x_*^2 = \sqrt{n}\sigma_*^2/c.$$

Plugging this into $\phi(\sigma)$ yields

$$\phi(\sigma_*) \le \Big(\frac{C}{c} + \frac{C}{c^2}\Big)\sqrt{n}\sigma_*^2 \le \sqrt{n}\sigma_*^2,$$

which implies that $\phi(\sigma_*)/\sigma_*^2 \leq \sqrt{n}$. And $\phi(x)/x$ and therefore $\phi(x)/x^2$ is non-increasing function, which conclude that $cx_*$ the upper bound for the solution of equation $\phi(\sigma)/\sigma^2 = \sqrt{n}$. Let the solution of $C(\sqrt{V}x + \sqrt{V}\int_0^x \log(1/\delta)d\delta) = \sqrt{n}x^2$ be $x_*$. Similarly,

$$\xi(x_*)/x_*^2 \leq C(\sqrt{V}x + \sqrt{V}\int_0^x \log(1/\delta)d\delta)/x^2 \leq \sqrt{n},$$

which implies $x_*$ is a upper bound of the solution of $\xi(x) = \sqrt{n}x^2$ with the non-increasing property of $\xi(x)/x^2$.

Define the non-increasing function $g(x) = \frac{\int_0^x \sqrt{\log(1/\epsilon)}d\epsilon}{x}$ with property $g(1) \approx 0.88 > 1/2$. Therefore, $g(x) > 1/2$ and $\frac{\sqrt{V}}{x} \leq 2 * \frac{\sqrt{V}g(x)}{x}$ for $x \in (0,1)$. With the same logic as in the last paragraph the solution of the equation of $C(\frac{\sqrt{V}g(x)}{x} + \frac{\sqrt{V}}{x}) = \sqrt{n}$ is upper bounded by that of $3C\frac{\sqrt{V}g(x)}{x} = \sqrt{n}$. Finally, the solution of equation $\phi(x) = \sqrt{n}x^2$ is upper bounded by the solutions

$$3C\frac{\sqrt{V}g(x)}{x} = \sqrt{n}.$$

For arbitrary $0 < \alpha < 1$, we have

$$\int_0^{x_*} \sqrt{\log 1/\epsilon}d\epsilon / x_*^2 = \frac{\int_0^{x_*}\sqrt{\log 1/\epsilon}d\epsilon}{x_*^{1-\alpha}}\frac{C}{x_*^{1+\alpha}} = \sqrt{\frac{n}{V}}.$$

It's easy to verify that $\lim_{x\to 0} \frac{\int_0^x \sqrt{\log(1/\epsilon)}d\epsilon}{x^{1-\alpha}} = 0$ and it's a bounded by some constant $C$. Therefore, we can bound $x_*$ by

$$\sqrt{\frac{n}{V}} = \frac{\int_0^{x_*}\sqrt{\log 1/\epsilon}d\epsilon}{x_*^{1-\alpha}}\frac{1}{x_*^{1+\alpha}} \leq \frac{C^2}{x_*^{1+\alpha}} \iff x_*^2 \leq \left(\frac{V}{n}\right)^{1-\frac{\alpha}{1+\alpha}} = \left(\frac{V}{n}\right)^{1-\gamma},$$

where $\gamma = \frac{\alpha}{1+\alpha}$ is an arbitrary number in $(0, 1/2)$.

Therefore, the upper bound of solution of $\phi(w(\epsilon_*)) = \phi(\epsilon_*) = \sqrt{n}\epsilon_*^2$ is

$$\epsilon_*^2 = C^2\left(\frac{V}{n}\right)^{1-\gamma},$$

where $\gamma$ is arbitrary number in $(0, 1/2)$.

**Step 4: Final result**

Combine all the results above and Theorem 8.3 in Massart (2003), we can conclude that with probability at least $1 - 4\delta$,

$$L(\hat{f}, \hat{t}) - L(f^*, t^*)$$
$$\leq \mathcal{O}\left(\left(\frac{V}{n}\right)^{1-\gamma} + \frac{V}{n} + \frac{\log(1/\delta)}{n} + 2\inf_{f\in\mathcal{F}_n, t\in\mathbb{R}}\{L(f,t) - L(f^*, t^*)\} + \frac{1 \wedge \left(\frac{V}{n}\right)^{1-\gamma}}{(\frac{V}{n})^{1-\gamma}}\frac{\log(1/\delta)}{n}\right)$$
$$\leq \mathcal{O}\left(\left(\frac{V}{n}\right)^{1-\gamma} + 2\inf_{f\in\mathcal{F}_n}\{L(f,t^*) - L(f^*,t^*)\} + \frac{\log(1/\delta)}{n}\right)$$
$$\leq \mathcal{O}\left(\left(\frac{V}{n}\right)^{1-\gamma} + \inf_{f\in\mathcal{F}_n}\|f - f^*\|_\infty^2 + \frac{\log(1/\delta)}{n}\right).$$

Note that the above bound holds for all $\gamma \in (0, 1/2)$ and

$$C_1\|\hat{f} - f^*\|_{L_2(P_X)}^2 \leq d_p^2((\hat{f}, \hat{t}), (f^*, t^*)) \leq L(\hat{f}, \hat{t}) - L(f^*, t^*)$$
$$\leq \mathcal{O}\left(\frac{V}{n}\left(1 + \frac{\log(1/\delta)}{n}\right) + \inf_{f\in\mathcal{F}}\|f(x) - f^*(x)\|_\infty^2\right). \tag{16}$$

**Proof of Corollary 4.7.** Given any arbitrary $N, L \in \mathbb{N}^+$, we assume that $\mathcal{F}_n$ is the deep ReLU networks $\mathcal{F}_n$ with width

$$\mathcal{N} = 3^{p+3} \max \left( p \lfloor N^{1/p} \rfloor, N+1 \right)$$

and depth $\mathcal{L} = 12L + 14 + 2p$. By Theorem 1.1 in Shen (2020), we have the approximation error

$$\inf_{f \in \mathcal{F}_n} \|f - f^*\|_\infty^2 = \mathcal{O} \left( (\mathcal{N}\mathcal{L})^{-\frac{4\beta}{p}} \right). \tag{17}$$

Since the ReLU-activated neural network class is a VC class, the parameter $V$ in assumption 4.5 can be its VC dimension (see Theorem 7 of Bartlett et al. (2019))

$$\mathcal{S}\mathcal{L} \log (\mathcal{S}). \tag{18}$$

Based on (17) and (18), the following holds with probability at least $1 - 4\delta$ for any $\delta > 0$ by Theorem 4.6,

$$L(\hat{f}, \hat{t}) - L(f^*, t^*) = \mathcal{O} \left( \frac{\mathcal{S}\mathcal{L} \log (\mathcal{S})}{n} + (\mathcal{N}\mathcal{L})^{-\frac{4\beta}{p}} + \frac{\mathcal{S}\mathcal{L} \log (\mathcal{S}) \log(1/\delta)}{n} \right).$$

Notice that, for any neural network $f \in \mathcal{F}_n$, its size satisfies

$$\mathcal{S} \leq \mathcal{N}(p+1) + (\mathcal{N}^2 + \mathcal{N})(\mathcal{L} - 1) + \mathcal{N} + 1 = \mathcal{O} (\mathcal{N}^2 \mathcal{L}).$$

Therefore,

$$L(\hat{f}, \hat{t}) - L(f^*, t^*)$$
$$= \mathcal{O} \left( \frac{\mathcal{N}^2 \mathcal{L}^2 \log (\mathcal{N}^2 \mathcal{L})}{n} + (\mathcal{N}\mathcal{L})^{-\frac{4\beta}{p}} + \frac{\mathcal{N}^2 \mathcal{L}^2 \log (\mathcal{N}^2 \mathcal{L}) \log(1/\delta)}{n} \right).$$

To achieve the balance between the first two terms, we consider the width-fixed ReLU neural network class $\mathcal{F}_n$ with $L = n^{\frac{p}{2(p+2\beta)}}$ and $N$ being any positive integers, which leads to

$$L(\hat{f}, \hat{t}) - L(f^*, t^*) = \mathcal{O} \left( n^{-\frac{2\beta}{p+2\beta}} \log n + n^{-\frac{2\beta}{p+2\beta}} \log n \log(1/\delta) \right)$$
$$= \mathcal{O} \left( n^{-\frac{2\beta}{p+2\beta}} \{ 1 + \log(1/\delta) \} \log n \right).$$

By equality (16), $\|\hat{f} - f^*\|_{L_2(P_X)}^2$ has the same convergence rate with $L(\hat{f}, \hat{t}) - L(f^*, t^*)$.

