# OpenReview forum: "Deep Principal Support Vector Machines for Nonlinear Sufficient Dimension Reduction"
_ICML.cc/2025/Conference — ICML 2025 poster_

### Official Review · Reviewer_dTHf · 2025-03-07

**Overall Recommendation:** 2

**Summary:**

The paper focuses on SDR replacing functions in RKHS with neural networks.
This is motivated by the potential advantages of neural networks in handling complex data structures. The authors theoretically demonstrate the unbiasedness of their method and provide a non-asymptotic upper bound for the estimation error, using principles of classification ensembles for nonlinear SDR. While the core idea is sound, the paper lacks a truly comprehensive and balanced approach, especially on the complexity and practical considerations, failing to show the true value of the proposal.

**Claims And Evidence:**

The paper claims that using neural networks offers advantages over traditional kernel methods in SDR. However, it doesn't provide a detailed analysis of the computational complexity vs. kernel methods. While it's generally stated that neural networks can be more efficient for large datasets, the specific computational costs are not comprehensively quantified and compared. Moreover, a comparison of the number of parameters should be done.

The paper's assertion of superior performance lacks substantial supporting evidence. Little empirical evidence might be the greatest weakness of the paper. The simulations and real data analysis are limited in scope and don't fully validate the claimed advantages, as the scope is very specific. The results presented, while showing some improvements, are not conclusive enough to definitively establish the superiority of the proposed method in a broader context.

**Essential References Not Discussed:**

N\A

**Experimental Designs Or Analyses:**

The experimental design is limited in its ability to fully support the claims made in the paper. The claim of outperformance needs qualification. The improvements over other methods in Table 1 are sometimes marginal, and the experiments are limited in scope (low dimensionality, specific datasets).

**Methods And Evaluation Criteria:**

The evaluation is limited by the scope of the datasets used. A wider range of benchmark datasets, especially higher-dimensional ones, and a more rigorous analysis of computational complexity would further strengthen the evaluation. The datasets utilized are relatively low-dimensional, and the performance on these might not generalize to more complex, high-dimensional data encountered in modern scientific applications. The lack of comparison with state of the art on high dimentional or more datasets limits the work. The lack of ablation study in order to understand the relative importance of different components in the network is an issue.

**Other Comments Or Suggestions:**

The paper could benefit from a more thorough investigation of the sensitivity of the proposed method to different hyperparameter settings. The current sensitivity analysis in the supplementary material is limited. Exploring a wider range of architectures and regularization techniques would provide a more complete picture of the method's robustness. Additionally, a discussion of potential strategies for mitigating the "black box" nature of neural networks in this context would be valuable. For instance, exploring techniques like attention mechanisms or feature importance analysis could enhance the interpretability of the learned representations. Finally, it would improve the presentation to include a figure illustrating the proposed neural network architecture.

**Other Strengths And Weaknesses:**

A significant weakness is the potential lack of interpretability inherent in using neural networks. Given that they utilize neural networks, interpretability can be reduced, and there should be some discussion around this. While neural networks can offer flexibility and representational power, they often come at the cost of being "black boxes." The paper doesn't address how the learned representations can be interpreted or understood in the context of the original data, which is crucial for many scientific applications where domain knowledge is important.

The impact statement is missing.

**Questions For Authors:**

How was the specific neural network architecture (number of layers, number of neurons per layer, activation functions) chosen, and what was the rationale behind this choice? Did you explore other architectures, and if so, how did their performance compare?

Could you elaborate on the hyperparameter tuning process? What range of values was considered for each hyperparameter, and how was the optimal configuration determined?

What are the practical limitations of the proposed method in terms of computational resources (training time, memory usage) when applied to very high-dimensional datasets?

How can the learned representations be interpreted in a way that provides meaningful insights into the relationship between the predictors and the response variable? Are there any plans to incorporate techniques to improve interpretability?

**Relation To Broader Scientific Literature:**

The relevant literature is discussed, placing the proposed method within the context of existing SDR techniques. The paper appropriately cites key works in the field of SDR, including both linear and nonlinear approaches. However, the discussion could be expanded to include a more critical comparison with a wider range of contemporary methods, particularly those that also leverage neural networks or deep learning for dimension reduction.

**Theoretical Claims:**

The theoretical claims regarding the unbiasedness of the optimal solution and the non-asymptotic upper bound for the estimation error seem mathematically OK.

---

> ### Author Rebuttal · Authors · 2025-03-31
>
> Thanks to the reviewer for helpful comments. We will refine the manuscript.
>
> > **Q1**. The paper doesn't provide a detailed analysis of the computational complexity vs. kernel methods.
>
> **A1**. A brief theoretical analysis of computational costs is provided in **Q4** of Reviewer 1. The cost of our methods is
> $$\mathcal{O}(hnt\mathcal{L} \max\\{p, \mathcal{N}\\}^2 + h^2n + h^3),$$  which is linear with $n$.
> Other deep methods avoid binarization, reducing it to
> $$ \mathcal{O}(nt\mathcal{L} \max\\{p, \mathcal{N}\\}^2).$$
>
> Kernel-based SDR methods, requiring the eigendecomposition of an $n \times n$ matrix, incur
> $$\mathcal{O}(n^3).$$
> While discretization increases computation, it enhances robustness for both regression and classification tasks, particularly for data with many outliers.
>
> >**Q2**. The experiments are limited in scope.
>
> **A2**. Thanks for your comments. Our focus is to provide a theoretical perspective on a new robust nonlinear SDR framework, especially considering the embedding of neural networks. Importantly, we establish a faster non-asymptotic convergence rate compared to Zheng et al. (2022) and Huang et al. (2024). Our rate nearly reaches the minimax rate of nonparametric regression.
> The existing experiments show results comparable to other approaches. More experiments like simulation with large $p=100$ is added in the revision.
>
> * Zheng, S., Lin, Y., & Huang, J. (2022). Deep Sufficient Representation Learning via Mutual Information. arXiv preprint.
> * Huang, J., Jiao, Y., Liao, X., Liu, J., & Yu, Z. (2024). Deep dimension reduction for supervised representation learning. IEEE Transactions on Information Theory.
>
> >**Q3**. Literature.
>
> **A3**. We have included the latest deep dimension reduction methods for comparison.
>
> >**Q4**. How was the specific neural network architecture chosen, and what was the rationale behind this choice? Did you explore other architectures?
>
> **A4**. This structure follows an expansion-contraction pattern, similar to a reverse bottleneck in autoencoders: widening layers map $x$ to a higher-dimensional space (akin to RKHS feature mapping), while narrowing layers extract a lower-dimensional representation. The default network is a feedforward ReLU network with hidden dimensions $2^{D_1}, 2^{D_1+1}, 2^{D_1}, 2^{D_1-1}, \dots, 16, 1$, where $D_1 = \lfloor \log_2 p \rfloor + 1$ and $p$ is the input dimension. Our method performs well with a simple feedforward network, so we did not explore more complex architectures.
>
> >**Q5**. Could you elaborate on the hyperparameter tuning process? What range of values was considered for each hyperparameter, and how was the optimal configuration determined?
>
> **A5**. The hyperparameter tuning process is primarily introduced in section A.3. Some hyperparameters, such as optimizer(Adam), batch size (100) and the number of epochs(100), remain unchanged, as they are less significant compared to the others: binarization count, $\lambda$, learning rate, and neural network structure. The range of them can be found in Table 2. Table 2 shows that increasing the binarization count improves performance, consistent with classical PSVM results. A smaller $\lambda$ also leads to better performance, as it shifts the focus toward dimension reduction rather than variance constraints. Notably, $\lambda$ is not a regularization parameter like in $L_2$-regularized regression. The ablation study on learning rate and network structure suggests that the default setting is optimal. The optimal configuration uses the default network, 10 binarizations to balance performance and computational cost, $\lambda = 0.01$, and a learning rate of 0.001, chosen by cross-validation.
>
> Following the reviewer's suggestion, we will conduct a more in-depth investigation into the sensitivity of the proposed method to different hyperparameters.
>
> >**Q6**. What are the practical limitations of the proposed method in terms of computational resources when applied to very high-dimensional datasets?
>
> **A6**. The theoretical computational cost, addressed in **A1**, scales quadratically with the input dimension $p$, which is consistent with most neural network-based methods. Consequently, the practical limitations of our approach are similar to those of typical deep learning methods.
>
> >**Q7**. How can the learned representations be interpreted in a way that provides meaningful insights into the relationship between the predictors and the response variable?
>
> **A7**. For classification tasks like MNIST, the 2D visualization in Figure 1 offers meaningful insights into the relationship between predictors and the responses. Additionally, we will include scatter plots where colors correspond to Y (e.g., brighter red for larger Y and light green for smaller Y). These visualizations will provide a clearer understanding of the learned representations. Additionally, we appreciate your suggestion—conducting feature importance analysis will further enhance the interpretability of our method.

---

### Official Review · Reviewer_Eu69 · 2025-03-09

**Overall Recommendation:** 2

**Summary:**

This paper introduce a unified framework for nonlinear sufficient dimension reduction method based on classification ensemble. The framework proposed in this paper almost includes kernel principal SVM, which is a nonlinear sufficient dimension reduction method using the reproducing kernel Hilbert space and SVM. Here, the neural network function class is considered to replace the reproducing kernel Hilbert space to implement a more flexible deep nonlinear sufficient dimension reduction. The authors demonstrate theoretical unbiasedness of the optimal solution of the population objective function and a non-asymmetric upper bound for the estimation error. All these results demonstrate considerable competitiveness of the newly proposed deep nonlinear sufficient dimension reduction method.

**Claims And Evidence:**

Yes they are.

**Essential References Not Discussed:**

No.

**Experimental Designs Or Analyses:**

Experiments are good

**Methods And Evaluation Criteria:**

Make sense.

**Other Comments Or Suggestions:**

N/A

**Other Strengths And Weaknesses:**

Strengths: Provides both theoretical and empirical soundness of the proposed method.

Weaknesses: In the Preliminary part Section 2.1, when the authors introduce the nonlinear dimension reduction problem, it would be better if there could be some concrete example, as this introduction seems very abstract.

**Questions For Authors:**

I wonder if the authors can present some concrete meanings of the Assumptions 4.2-4.5 listed in the Section 4 Theoretical Results. They seem to be very theoretical.

**Relation To Broader Scientific Literature:**

N/A

**Theoretical Claims:**

Yes. They are solid.

---

> ### Author Rebuttal · Authors · 2025-03-30
>
> Thank you for acknowledging our theoretical contribution. We will address your concerns in the revision and below are our detailed responses.
>
> > **Q1**. In the Preliminary part Section 2.1, when the authors introduce the nonlinear dimension reduction problem, it would be better if there could be some concrete example, as this introduction seems very abstract.
>
> **A1**. To illustrate the motivation behind sufficient dimension reduction (SDR), we present two simple examples.
> ### Example 1: Classification Problem
> Consider the classification problem:
> $$
> Y = \mathbb{1} \left( X_1^2 + X_2^2 - 1 + \varepsilon > 0 \right),
> $$
> where  $X_1, X_2 \sim \mathcal{N}(0,1)$ and $\varepsilon \sim \mathcal{N}(0, 0.2)$.  Then, the normalized SDR function is given by
> $$  f_0 (X) = \frac{X_1^2 + X_2^2}{2} - 1.  $$
> Here, a linear transformation is applied to ensure $E\big(f_0 (X)\big)=0，\operatorname{Var}\big(f_0 (X)\big)=1$.
> ### Example 2: Multivariate Regression Problem
> Now, consider a multivariate regression problem where $Y \in \mathbb{R}^2$:
> $$  Y_1 = \sin(X_1 + X_2^2) + \varepsilon_1, \quad Y_2 = \cos(X_3 X_4) + \varepsilon_2,  $$
> where $X_1, X_2, X_3, X_4 \sim \mathcal{N}(0,1)$ and $\varepsilon_1, \varepsilon_2 \sim \mathcal{N}(0, 0.2)$.  In this case, the normalized SDR function is  $$  \boldsymbol{f}_0(X) = (a_0 \sin(X_1 + X_2^2) + b_0, a_1\cos(X_3 X_4) + b_1),  $$
> where $a_i$ and $b_i, i=1,2$, are introduced to ensure $E\big(\boldsymbol{f}_0 (X)\big)=0，\operatorname{Var}\big(\boldsymbol{f}_0(X)\big)=I_2$.
> ### General Formulation for Regression
> More generally, we consider the model
> $$
> Y = \boldsymbol{g}(\boldsymbol{f}_0(X)) + \boldsymbol{\epsilon},
> $$
> where $\boldsymbol{f}_0: \mathbb{R}^p \to \mathbb{R}^d$ and $\boldsymbol{g}: \mathbb{R}^d \to \mathbb{R}^q$. Our goal is to identify the low-dimensional nonlinear representation $\boldsymbol{f}_0(X)$.  In this general setting, traditional linear sufficient dimension reduction breaks down.
>
> >**Q2**. I wonder if the authors can present some concrete meanings of Assumptions 4.2-4.5 listed in Section 4 Theoretical Results. They seem to be very theoretical.
>
> **A2**. **Assumption 4.2** is a standard boundedness assumption commonly used in statistical learning theory. It states that both the ground-truth function and the neural network are bounded.
>
> **Assumption 4.3** assumes that $f^*$ is $\beta$-Hölder smooth. A function $f: [0,1]^p \to \mathbb{R}$ is $\beta$-Hölder continuous if there exist $\beta \in (0,1]$ and $\zeta > 0$ such that
>
> $$
> \left| f(x_1) - f(x_2) \right| \leq \zeta \|x_1 - x_2\|_2^{\beta}, \quad \text{for any} \quad x_1, x_2 \in [0,1]^p.
> $$
>
> Since $\|x_1 - x_2\|_2^{\beta_1} \geq \|x_1 - x_2\|_2^{\beta_2}$ for $\beta_1 \leq \beta_2$ and $\|x_1 - x_2\|_2 \leq 1$, it follows that functions with smaller $\beta$ exhibit more pronounced variations in local regions. In other words, the parameter $\beta$ quantifies the smoothness or regularity of the function:
>
> - A higher $\beta$ corresponds to a smoother function with fewer abrupt changes.
> - A lower $\beta$ indicates a less smooth, more irregular function with greater variation.
>
> The smoothness level $\beta$ determines how easily the function can be approximated, with larger $\beta$ indicating easier approximation. Finally, $\beta$ and the input dimension $d$ jointly determine the convergence rate of our estimator, please see Corollary 4.7.
>
> **Assumption 4.4** requires that the convex loss function is Lipschitz continuous, a condition satisfied by the hinge loss.
>
> **Assumption 4.5** states that
>
> $$\log N\left(\epsilon, \mathcal{F_n},\|\cdot\|_{L_2(Q)}\right) \leq C V \left(1+\log (1 / \epsilon)\right). $$
>
> Here, $V$ can be regarded as the VC dimension of a certain VC class $\mathcal{F}_n$, which always satisfies this assumption. For example, the VC dimension of the neural network class is determined by parameters such as the width and depth of the network. The VC dimension is a measure of the size (capacity, complexity, or expressive power) of a hypothesis class $\mathcal{F_n}$:
>
> - A higher VC dimension indicates a more powerful hypothesis class, which can lead to overfitting.
> - A lower VC dimension reduces flexibility but may lead to underfitting.
>
> Hence the performance of our estimator is relative to the trade-off between statistical error $V/n$ and approximation error $\inf_{f \in \mathcal{F_n}} \|f - f^*\|_{\infty}^2$, both affected by the complexity $V$ of $\mathcal{F_n}$:
>
> - A larger, more complex class $\mathcal{F}_n$ increases $V$, leading to lower approximation error but higher statistical error.
> - A smaller, simpler class $\mathcal{F_n}$ decreases $V$, reducing statistical error but potentially increasing approximation error.
>
> The result of the trade-off is formalized in Corollary 4.7.

---

### Official Review · Reviewer_RtJT · 2025-03-13

**Overall Recommendation:** 3

**Summary:**

The paper introduces a general framework of nonlinear sufficient dimension reduction (SDR) based on the previous works of principal support vector machine (PSVM). Classical algorithms such as linear SDR and kernel PSVM are special cases of this new framework. When applying deep neural network to this framework, we obtain the deep principal support vector machine (DPSVM) algorithm.
This paper also provides theoretical guarantees for the effectiveness of this new framework and the DPSVM algorithm, including the conditions for the unbiasedness of the optimal solutions, and the minimax convergence rate for the mean square estimation error under Holder assumption when applying deep neural network to this framework.

**Claims And Evidence:**

The statements of this paper are clear. The new algorithm introduced by this paper is based on a series of previous works, and a number of references and experimental data are provided.

**Essential References Not Discussed:**

None.

**Experimental Designs Or Analyses:**

The paper provides adequate experimental results for the comparison between DPSVM and other nonlinear SDR algorithms on several artificial datasets such as MNIST, demonstrating impressive competitiveness of the new algorithm.
As a potential limitation of the experiment part, section 6 mentioned that one of the limitations of DPSVM is the speed, while this paper does not provide experimental results to compare the speed of DPSVM with those of other algorithms.

**Methods And Evaluation Criteria:**

The nonlinear SDR framework introduced in this paper has the following strengths:
1. It is very flexible and can be applied to various types of model spaces, including reproducing kernel Hilbert spaces and deep neural network spaces.
2. In contrast with many previous works on PSVM, the new framework allow the response variable to be multi-dimensional, implying that it is suitable to more various types of regression and classification tasks.
There are a number of limitations in this paper:
1. The convergence results (Theorem 4.6 and Corollary 4.7) only analyze the properties of the empirical loss minimizer, while does not put the network training into consideration. Since the training speed may be one of the limitations of DPSVM, the analysis of the network optimization is a critical issue for DPSVM.
2. The structure dimension estimation discussed in section 3.2 lacks further theoretical discussion;
3. The DPSVM algorithm requires the binary discretization of the response variable. For continuous response variable, the error arising from discretization is not discussed in this paper.

**Other Comments Or Suggestions:**

None.

**Other Strengths And Weaknesses:**

None.

**Questions For Authors:**

The DPSVM algorithm discussed in this paper requires the discretization of the response variable, which may cause deceleration of training speed. Is there any nonlinear SDR algorithms that can directly deal with continuous response? If it exists, how is the comparison between it and DPSVM?

**Relation To Broader Scientific Literature:**

None.

**Theoretical Claims:**

We have checked the proofs and found no evident mistakes.

---

> ### Author Rebuttal · Authors · 2025-03-28
>
> Thank you for the insightful feedback. Below are our responses.
>
> > **Q1** The convergence results (Theorem 4.6 and Corollary 4.7) only analyze the properties of the empirical loss minimizer, while does not put the network training into consideration. Since the training speed may be one of the limitations of DPSVM, the analysis of the network optimization is a critical issue for DPSVM.
>
> **A1**. Thank you for your valuable opinion. The training loss or optimization error is often overlooked in the statistical literature. For instance, Schmidt-Hieber (2020) establishes a theoretical bound for deep neural networks in nonparametric regression, where he defines the gap between the estimator derived from training and the true global empirical minimizer as $\Delta_n$. However, he does not provide any theorems specifically addressing this term, instead treating it as either negligible or dominated by statistical error.
>
> The analysis of the training process in neural networks is indeed complex and demands considerable effort. Some recent works, such as Jentzen & Welti (2023) and Beck, Jentzen & Kuckuck (2022), incorporate training error into their analysis of least squares regression. However, as noted by the authors, the resulting convergence rates are far from optimal and are hindered by the curse of dimensionality. While we could adopt a similar decomposition—including statistical, optimization, and approximation errors—such an approach may obscure the core theoretical contribution of our work.
>
> * Schmidt-Hieber, J. (2020). Nonparametric regression using deep neural networks with ReLU activation function. The Annals of Statistics, 48(4), 1875–1897.
>
> * Beck, C., Jentzen, A., & Kuckuck, B. (2022). Full error analysis for the training of deep neural networks. Infinite Dimensional Analysis, Quantum Probability and Related Topics, 25(02), 2150020.
>
> * Jentzen, A., & Welti, T. (2023). Overall error analysis for the training of deep neural networks via stochastic gradient descent with random initialization. Applied Mathematics and Computation, 455, 127907.
>
> >**Q2**  The structure dimension estimation discussed in section 3.2 lacks further theoretical discussion.
>
> **A2**. Yes, you are correct. Structure dimension estimation is inspired by traditional linear sufficient dimension reduction (SDR) methods such as SIR (Li, 1991). Here, we follow the ladle method from Luo & Li (2016)  and construct a positive-definite matrix, $E_n (\widehat{\mathbf{f}}^\intercal(X)\widehat{\mathbf{f}}(X))$, which converges to $E ({\mathbf{f}}^\intercal(X){\mathbf{f}}(X))$.  Luo & Li (2016) have demonstrated that the ladle method consistently selects the true structure dimension as $n$ approaches infinity.  We have used this method in artificial datasets, and will subsequently add simulation experiments to verify the effectiveness of structure dimension estimation.
>
> * Li, K.-C. (1991). Sliced inverse regression for dimension reduction. Journal of the American Statistical Association, 86(414), 316–327.
>
> * Luo, W., & Li, B. (2016). Combining eigenvalues and variation of eigenvectors for order determination. Biometrika, 103(4), 875–887.
>
> >**Q3**. The DPSVM algorithm requires the binary discretization of the response variable. For the continuous response variable, the error arising from discretization is not discussed in this paper.
>
> **A3**. If we discretize $Y$ into $\tilde{Y}$, our goal is to find $f$ such that $\tilde{Y} \perp X \mid f(X)$.  It is known that
> $$Y \perp X \mid f(X) \implies \tilde{Y} \perp X \mid f(X),$$
> but the reverse does not necessarily hold. However, both functions $f$ belong to the central class, which is fundamental for sufficient dimension reduction (SDR).  In nonlinear SDR, the central class forms an infinite-dimensional space when $X$ is real-valued. Since fully recovering this space is infeasible, we can only approximate it. Therefore, the error introduced by discretization isn't a big issue here.
>
> >**Q4**. This paper does not provide experimental results to compare the speed of DPSVM with those of other algorithms.
>
> **A4**. Theoretical computational costs can be derived, but for brevity, we present only the final result.
> Let $n, p, \mathcal{L}, \mathcal{N}$ be the sample size, input dimension, number of hidden layers, and maximum layer width, respectively. With batch size $b$ and epochs $t$, our method has a total cost of
> $$\mathcal{O}(hnt\mathcal{L} \max\\{p, \mathcal{N}\\}^2 + h^2n + h^3),$$
> where $\mathcal{O}(nt\mathcal{L} \max\\{p, \mathcal{N}\\}^2)$ accounts for training per binarization, $\mathcal{O}(h^2n)$ arises from matrix multiplication in $E_n (\widehat{\mathbf{f}}^\intercal(X)\widehat{\mathbf{f}}(X))$, and $\mathcal{O}(h^3)$ is due to eigendecomposition of an $h \times h$ matrix.
> In contrast, deep methods without binarization incur
> $$\mathcal{O}(nt\mathcal{L} \max\\{p, \mathcal{N}\\}^2).$$
> We will add relevant experiments and theoretical complexity analysis in the revision.

---

> > ### Comment · Reviewer_RtJT · 2025-04-04
> >
> > Thank you for your response. I will keep the current score

---

> > > ### Author Response · Authors · 2025-04-06
> > >
> > > We sincerely appreciate your positive evaluation of our work.

---

### Decision · Program_Chairs · 2025-05-01

**Decision:**

Accept (poster)

**Comment:**

This paper introduced a unified framework for nonlinear sufficient dimension reduction based on classification ensemble. Under the proposed framework, kernel principal SVM is extended by replacing kernel function with neural network to implement a more flexible deep nonlinear sufficient dimension reduction method. Reviewers consistently praised the soundness of theoretical results and competitive experimental results. However, reviewers also raised some key concerns including the lack of empirical computation complexity analysis and the datasets used are not high-dimensional, so the applications of the proposed method to real datasets can be limited. The authors’ responses try to clarify some of these concerns such as providing theoretical computation analysis of the proposed method instead of empirical comparisons, but not satisfactory especially for the concern of high-dimensional data (p=100 simulation data is not considered as high-dimensional). As a result, weak accept is recommended.